# Long-Term Fairness with Unknown Dynamics

**Tongxin Yin**[*]
Electrical and Computer Engineering
University of Michigan
Ann Arbor, MI 48109
tyin@umich.edu

**Reilly Raab**[*]
Computer Science and Engineering
University of California, Santa Cruz
Santa Cruz, CA 95064
reilly@ucsc.edu

**Mingyan Liu**
Electrical and Computer Engineering
University of Michigan
Ann Arbor, MI 48109
mingyan@umich.edu

**Yang Liu**
University of California, Santa Cruz
ByteDance Research
Santa Cruz, CA 95064
yangliu@ucsc.edu

## Abstract

While machine learning can myopically reinforce social inequalities, it may also be used to dynamically seek equitable outcomes. In this paper, we formalize long-term fairness as an online reinforcement learning problem for a policy affecting human populations. This formulation accommodates dynamical control objectives, such as achieving equitable population *states*, that cannot be incorporated into static formulations of fairness. We demonstrate that algorithmic solutions to the proposed fairness problem can adapt to unknown dynamics and, by sacrificing short-term incentives, drive the policy-population system towards more desirable equilibria. For the proposed setting, we develop an algorithm that adapts recent work in online learning and prove that this algorithm achieves simultaneous probabilistic bounds on cumulative loss and cumulative violations of fairness. In the classification setting subject to group fairness, we compare our proposed algorithm to several baselines, including the repeated retraining of myopic or distributionally robust classifiers, and to a deep reinforcement learning algorithm that lacks fairness guarantees. Our experiments model human populations according to evolutionary game theory and integrate real-world datasets.

## 1 Introduction

As machine learning (ML) algorithms are deployed for tasks with real-world social consequences (e.g., school admissions, loan approval, medical interventions, etc.), the possibility exists for runaway social inequalities (Crawford and Calo, 2016; Chaney et al., 2018; Fuster et al., 2018; Ensign et al., 2018). While "fairness" has become a salient ethical concern in contemporary research, the closed-loop dynamics of real-world systems with feedback, i.e., comprising ML-driven decisions and populations that mutually adapt to each other (Fig. 5 in the supplementary material) remain poorly understood. We assert that realistic scenarios are often described by fundamentally **unknown dynamics**: Even with models of human behavior based on rational behavior or evolutionary game theory, utilities and risk-preferences are generally unknown or uncertain. For this reason, we advocate for dynamics-agnostic formulations of fairness, for which reinforcement learning is a natural fit.

In this paper, our **primary contribution** is to consider the problem of *long-term fairness*, or algorithmic fairness in the context of a dynamically responsive population, as a reinforcement learning (RL)

---

[*]These authors contributed equally to this work.

37th Conference on Neural Information Processing Systems (NeurIPS 2023).

problem subject to constraint. In our formulation, the central learning task is to develop a policy that minimizes cumulative loss (e.g., financial risk, negative educational outcomes, misdiagnoses, etc.) incurred by an ML agent interacting with a human population up to a finite time horizon, subject to constraints on cumulative "violations of fairness", which we refer to in a single time step as *disparity* and cumulatively as *distortion*.

Our central hypothesis is that an RL formulation of long-term fairness may allow an agent to learn to **sacrifice short-term utility in order to drive the system towards more desirable equilibria**. The core practical difficulties posed by our general problem formulation, however, are the potentially unknown dynamics of the system under control, which must be determined by the RL agent *online* (i.e., during actual deployment), and the general non-convexity of the losses or constraints considered. Additionally, our problem setting generally requires continuous state and action spaces for the RL agent, which preclude familiar methods that yield performance guarantees in discrete settings.

Our **secondary contributions** are 1) to show that long-term fairness can be solved within asymptotic, probabilistic bounds under certain dynamical assumptions and 2) to demonstrate that the problem can also be addressed more flexibly with existing RL methods. For theoretical guarantees, we develop `L-UCBFair`, an online RL method, and prove sublinear bounds on regret (suboptimiality of cumulative loss) and distortion (suboptimality of cumulative disparity) with high probability (Section 3.1). To demonstrate practical solutions without strong safety guarantees, we apply `R-TD3`, an adaptation of a well-known deep reinforcement learning method (viz., TD3) to a time-dependent Lagrangian relaxation of the central problem. We compare `L-UCBFair` and `R-TD3` to several baselines (Section 3.3), including myopic policies, in interaction with simulated populations initialized with synthetic or real-world data and updated according to evolutionary game theory (Section 4).

Finally, this paper considers fairness in terms of statistical regularities across (ideally) socioculturally meaningful *groups*. While internal conflict exists between different statistical measures of fairness (Corbett-Davies and Goel, 2018), we show that an RL approach to long-term fairness can mitigate trade-offs between immediately-fair policy decision *outcomes* (e.g., acceptance rate disparities (Dwork et al., 2012; Zemel et al., 2013; Feldman et al., 2015)) and causally-fair population states (e.g., qualification rate (Raab and Liu, 2021; Zhang et al., 2020)). In particular, we show that our proposed solutions can learn to avoid the familiar trap of pursuing short-term fairness metrics only to widen underlying disparities that demand escalating interventions at the expense of utility (Section 5.2).

## 1.1   Related Work

Our formulation of long-term fairness bridges recent work on "fairness in machine learning", which has developed in response to the proliferation of data-driven methods in society, and "safe reinforcement learning", which seeks theoretical safety guarantees in the control of uncertain dynamical systems.

**Dynamics of Fairness in Machine Learning**   We distinguish long-term fairness from the dynamics of fair allocation problems (Joseph et al., 2016; Jabbari et al., 2017; Tang et al., 2021; Liu et al., 2017) and emphasize side-effects of algorithmic decisions affecting future decision problems. By formalizing long-term fairness in terms of cumulative losses and disparities, we iterate on a developing research trend that accounts for the dynamical response of a human population to deployed algorithmic prediction: both as a singular reaction (Liu et al., 2018; Hu et al., 2019; Hu and Zhang, 2022) or as a sequence of mutual updates between the population and the algorithm (Coate and Loury, 1993; D'Amour et al., 2020; Zhang et al., 2020; Heidari et al., 2019; Wen et al., 2019; Liu et al., 2020; Hu and Chen, 2018; Mouzannar et al., 2019; Williams and Kolter, 2019; Raab and Liu, 2021).

Several prior works have considered the long-term fairness implications in the context of performative stability or optimality (Perdomo et al., 2020) with known, stateless dynamical transitions: Hu and Zhang (2022) characterize the convergence of systems typified by sequential, myopic policies while Hashimoto et al. (2018) contrast myopic policy with a modified objective satisfying distributional robustness. While Mouzannar et al. (2019) and Raab and Liu (2021) address stateful dynamical transitions, the cited works only treat myopic classifiers that optimize immediate utility (subject to fairness constraints) rather than learning to anticipate dynamical population responses. Finally, while Wen et al. (2019) explore reinforcement learning for long-term fairness, the discussion is limited to a tabular explore-then-commit approach over discrete state and action spaces. This approach does not

generalize to continuous spaces, where we provide separate and tighter bounds for both utility and disparity.

**Safe Reinforcement Learning** `L-UCBFair` furthers recent efforts in safe RL. While *model-based* approaches, in which the algorithm learns an explicit dynamical model of the environment, constitute one thread of prior work (Efroni et al., 2020; Singh et al., 2020; Brantley et al., 2020; Zheng and Ratliff, 2020; Kalagarla et al., 2021; Liu et al., 2021; Ding et al., 2021), leading algorithms in this domain are characterized by significant time and space complexity. Among *model-free* algorithms, the unknown dynamics of our setting preclude the use of a simulator that allows algorithms to search over arbitrary state-action pairs (Xu et al., 2021; Ding et al., 2020; Bai et al., 2022).

While Wei et al. (2022) introduce a model-free, simulator-free algorithm, the tabular approach they consider is only applicable to discrete state and action spaces, while our setting requires continuous state and action spaces. To tackle continuous *state* space, Ding et al. (2021) and Ghosh et al. (2022) consider linear dynamics: Ding et al. (2021) develop a primal-dual algorithm with safe exploration, and Ghosh et al. (2022) use a softmax policy design. Both algorithms are based on the work of Jin et al. (2020), which proposed a least squares value iteration (LSVI) method, using an Upper Confidence Bound (UCB) (Auer et al., 2002) to estimate a state-action "$Q$" function. In addition to continuous state spaces, `L-UCBFair` also addresses continuous *action* spaces. To our knowledge, `L-UCBFair` is the first model-free, simulator-free RL algorithm to provides theoretical safety guarantees for both discrete and **continuous state *and* action spaces**. Moreover, `L-UCBFair` achieves bounds on regret and distortion as tight as any algorithm thus far on discrete action space (Ghosh et al., 2022).

## 2 Problem Formulation

We motivate our formulation of long-term fairness with a binary classification task, though the general formulation we propose is more widely applicable. Given this initial task, we sequentially incorporate *fairness constraints*, then *population dynamics*, and then *cumulative loss and disparity*, before fully formalizing the central problem of optimizing cumulative loss subject to expected dynamics and constraints on cumulative disparity.

For our initial binary classification task, suppose that a random individual, sampled i.i.d. from a fixed population, has *features* $X \in \mathbf{R}^d$, a *label* $Y \in \{-1, 1\}$, and a demographic *group* $G \in \mathcal{G}$ (where $\mathcal{G} = [n]$ for $n \geq 2$). Denote the joint distribution of these variables in the population as $s := \Pr(X, Y, G)$ such that $X, Y, G \sim s$. The task is to predict $Y$ (as $\hat{Y}$) from $X$ and $G$ by choosing a classifier $a$, where $\hat{Y} \sim a(X, G)$, to minimize the empirical loss $\mathscr{L}(s, a) \stackrel{e.g.}{=} \mathrm{E}\left[L(Y, \hat{Y})\right]$, where $L$ denotes some individual loss such as zero-one-loss. In general, we allow arbitrary, (unit-interval) bounded loss functions $\mathscr{L}(s, a) \in [0, 1]$ such that the **basic classification task** is $\min_a \mathscr{L}(s, a)$.

The **standard, "fair" classification task** (*without* a dynamically responsive population) incorporates constraints on the choice of classifier $a$, such that the *disparity* $\mathscr{D} \in [0, 1]$ induced on distribution $s$ by $a$ is bounded by some value $c \in [0, 1]$. Formally, the task is $\min_a \mathscr{L}(s, a)$ subject to $\mathscr{D}(s, a) \leq c$. A standard measure of disparity $\mathscr{D}$ is the violation of "demographic parity" (Dwork et al., 2012). For example, when $\mathcal{G} = \{g_1, g_2\}$, $\mathscr{D}(s, a) \stackrel{e.g.}{=} \left|\xi_{s,a}(g_1) - \xi_{s,a}(g_2)\right|^2$, where $\xi_{s,a}(g) := \Pr(\hat{Y}=1 \mid G=g)$.

In this paper, we also wish to consider measures of fairness inherent to the distribution $s$ (e.g., parity of group *qualification* rates $\Pr(Y=1 \mid G=g)$). Such measures of fairness can only be affected by changes to $s$ and thus require dynamics, which are not captured by the above formulation (Raab and Liu, 2021; Zhang et al., 2020). We will see that such notions of disparity are well-suited to an RL formulation of long-term fairness.

**State, action, and policy** Adopting the semantics of *sequence* of dependent classification tasks, we identify the time-dependent distribution $s_\tau \in \mathcal{S}$ of individuals in the population as a *state* and the chosen classifier $a_\tau \in \mathcal{A}$ as an *action* of some algorithmic *agent* at time $\tau$. While state space $\mathcal{S}$ is arbitrary, we assume that action space $\mathcal{A}$ admits a Euclidean metric, under which it is closed (i.e., $\mathcal{A}$ is isomorphic to $[0, 1]^m, m \in \mathbf{Z}_{>0}$). We specify that $a_\tau$ is sampled stochastically according to the agent's current *policy* $\pi_\tau$, that is, $a_\tau \sim \pi_\tau(s_\tau)$. Additionally, we assume $s_\tau$ is fully observable at time $\tau \in \{1, 2, ...\}$. In practice, $s_\tau$ must be approximated from finitely many empirical samples, though this caveat introduces well-understood errors that vanish in the limit of infinitely many samples.

**Dynamics**  Moving beyond the one-shot "fair" classification task above, let us now assume that a population may react to classification, inducing the distribution $s$ to change. In practice, such "distribution shift" is a well-known, real-world phenomenon that can increase realized loss and disparity when classifiers are fixed (Chen et al., 2022). For classifiers that free to update in response to a mutating distribution $s$, subsequent classification tasks depend on the (stochastic) predictions made in previous tasks. In our formulation, we assume the existence of dynamical kernel $\mathbf{P}$ that maps a state $s$ and action $a$ at time $\tau$ to a *distribution over* possible states at time $\tau + 1$. That is, $s_{\tau+1} \sim \mathbf{P}(s_\tau, a_\tau)$.

We stipulate that $\mathbf{P}$ may be initially unknown, but we assume that it does not explicitly depend on time and may be reasonably approximated or learned "online". While real-world dynamics may depend on information other than the current distribution of classification-specific variables $\Pr(X, Y, G)$ — e.g., exogenous parameters, history, or additional variables of state — we have identified the dynamical state $s$ with the current, fully-observed distribution for simplicity and tractability.

**Reward and utility, value and quality functions**  Because the standard RL convention is to *maximize reward* rather than *minimize loss*, for an RL agent, we define the instantaneous reward $r(s_\tau, a_\tau) := 1 - \mathscr{L}(s_\tau, a_\tau) \in [0, 1]$ and a separate "utility" $g(s_\tau, a_\tau) := 1 - \mathscr{D}(s_\tau, a_\tau) \in [0, 1]$, where $r$ and $g$ do not explicitly depend on time $\tau$. We next propose to optimize anticipated *cumulative* reward subject to constraints on anticipated *cumulative* utility. Let the index $j$ represent either reward $r$ or utility $g$. We use the letter $V$ (for "value") to denote the future expected accumulation of $j$ over steps $[h, ..., H]$ (without time-discounting) starting from state $s$, using policy $\pi$. Likewise, we denote the "quality" of an action $a$ in initial state $s$ with the letter $Q$. For $j \in \{r, g\}$, we define

$$V_{j,h}^\pi(s) := \mathrm{E}\left[\sum_{\tau=h}^{H} j(s_\tau, a_\tau) | s_h = s\right] \quad ; \quad Q_{j,h}^\pi(s,a) := \mathrm{E}\left[\sum_{\tau=h}^{H} j(s_\tau, a_\tau)) \mid s_h = s, a_h = a\right].$$

Note that $V$ and $Q$ marginalize over the stochasticity of state transitions $s_{\tau+1} \sim \mathbf{P}(s_\tau, a_\tau)$ and the sampling of actions $a_\tau \sim \pi_\tau(s_\tau)$. By the boundedness of $r$ and, $g$, the values of $V$ and $Q$ belong to the interval $[0, H - h + 1]$.

**"Long-term fairness" via reinforcement learning**  The central problem explored in this paper is

$$\max_\pi \quad V_{r,1}^\pi(s) \quad \text{subject to} \quad V_{g,1}^\pi(s) \geq \tilde{c}. \tag{1}$$

We emphasize that this construction considers a finite time horizon of $H$ steps, and we denote the optimal value of $\pi$ as $\pi^\star$. We first assume that a solution to the problem exists bounded within the interior of the feasible set (i.e., *strict primal feasibility* or *Slater's condition*), as in Efroni et al. (2020), Ding et al. (2021), and Ghosh et al. (2022):

**Assumption 2.1** (Slater's Condition). $\exists \gamma > 0, \bar{\pi}$, such that $V_{g,1}^{\bar{\pi}}(s) \geq \tilde{c} + \gamma$.

**The Online Setting**  Given initially unknown dynamics, we formulate long-term fairness for the *online* setting (in which learning is only possible through actual "live" deployment of policy, rather than through simulation). As it is not possible to unconditionally guarantee constraint satisfaction in Eq. (1) over a finite number of online steps, we instead measure two types of *regret*: the suboptimality of a policy with respect to cumulative incurred loss, denoted $\mathrm{Regret}$, and the suboptimality of a policy with respect to cumulative induced disparity, denoted denoted $\mathrm{Drtn}$ for "distortion".

$$\mathrm{Regret}(\pi, s_1) := V_{r,1}^{\pi^*}(s_1) - V_{r,1}^\pi(s_1) \quad ; \quad \mathrm{Drtn}(\pi, s_1) := \max\left[0, \tilde{c} - V_{g,1}^\pi(s_1)\right]. \tag{2}$$

## 3   Algorithms and Analysis

We show that it is possible to provide guarantees for long-term fairness in the online setting. To do this, we develop `L-UCBFair`, the first model-free, simulator-free algorithm to provide such guarantees with continuous state and action spaces. Specifically, we prove probabilistic, sublinear bounds for regret and distortion under a linear MDP assumption (Assumption 3.1) and properly chosen parameters (Appendix B.1, in the supplementary material). We defer proofs to the supplementary material.

### 3.1 `L-UCBFair`

**Episodic MDP**  `L-UCBFair` inherits from a family of algorithms that treat an episodic Markov decision process (MDP) Jin et al. (2020). Therefore, we first map the long-term fairness problem to the episodic MDP setting, which we denote as MDP$(\mathcal{S}, \mathcal{A}, H, \mathbf{P}, \mathscr{L}, \mathscr{D})$. The algorithm runs for $K$ *episodes*, each consisting of $H$ time steps. At the beginning of each episode, which we index with $k$, the agent commits to a sequence of policies $\pi^k = (\pi_1^k, \pi_2^k, ..., \pi_H^k)$ for the next $H$ steps. At each step $h$ within an episode, an action $a_h^k \in \mathcal{A}$ is sampled according to policy $\pi_h^k$, then the state $s_{h+1}^k \in \mathcal{S}$ is sampled according to the transition kernel $\mathbf{P}(s_h^k, a_h^k)$. $s_1^k$ is sampled arbitrarily for each episode.

**Episodic Regret**  Because `L-UCBFair` predetermines its policy for an entire episode, we amend our definitions of regret and distortion over all $HK$ time steps by breaking them into a sum over $K$ episodes of length $H$:

$$\text{Regret}(K) = \sum_{k=1}^{K} \left( V_{r,1}^{\pi^*} \left( s_1^k \right) - V_{r,1}^{\pi^k} \left( s_1^k \right) \right) \quad ; \quad \text{Drtn}(K) = \max \left[ 0, \sum_{k=1}^{K} \left( \tilde{c} - V_{g,1}^{\pi^k} \left( s_1^k \right) \right) \right]. \quad (3)$$

**The Lagrangian**  Consider the Lagrangian $\mathcal{L}$ associated with Eq. (1), with dual variable $\nu \geq 0$:

$$\mathcal{L}(\pi, \nu) := V_{r,1}^{\pi} \left( s \right) + \nu \left( V_{g,1}^{\pi} \left( s \right) - \tilde{c} \right). \quad (4)$$

`L-UCBFair` approximately solves the primal problem, $\max_\pi \min_\nu \mathcal{L}(\pi, \nu)$, which is non-trivial, since the objective function is seldom concave in practical parameterizations of $\pi$. Let $\nu^*$ denote the optimal value of $\nu$; `L-UCBFair` assumes bound $\nu^* \leq \mathscr{V}$, given parameter $\mathscr{V}$, as described in the supplementary material (Assumption B.1).

**Assumption 3.1** (Linear MDP).  We assume MDP$(\mathcal{S}, \mathcal{A}, H, \mathbf{P}, \mathscr{L}, \mathscr{D})$ is a linear MDP with feature map $\phi : \mathcal{S} \times \mathcal{A} \to \mathbb{R}^d$. That is, for any $h$, there exist $d$ signed measures $\mu_h = \left\{ \mu_h^1, \ldots, \mu_h^d \right\}$ over $\mathcal{S}$ and vectors $\theta_{r,h}, \theta_{g,h} \in \mathbb{R}^d$ such that, for any $(s, a, s') \in \mathcal{S} \times \mathcal{A} \times \mathcal{S}$,

$$\mathbb{P}_h \left( s' \mid s, a \right) = \langle \phi(s, a), \mu_h \left( s' \right) \rangle; \quad r \left( s, a \right) = \langle \phi(s, a), \theta_{r,h} \rangle; \quad g \left( s, a \right) = \langle \phi(s, a), \theta_{g,h} \rangle.$$

Assumption 3.1 addresses the curse of dimensionality when state space $\mathcal{S}$ is the space of distributions over $X, Y, G$. This assumption is also used by Jin et al. (2020) and Ghosh et al. (2022), and a similar assumption is made by Ding et al. (2021). This assumption is well-justified in continuous time if we allow for infinite-dimensional Hilbert spaces Brunton et al. (2021), but in practice we require limited dimensionality $d$ for the codomain of $\phi$. In our experiments, we use a neural network to estimate a feature map $\hat{\phi}$ offline which approximately satisfies the linear MDP assumption (Appendix D.1).

#### 3.1.1  Construction

`L-UCBFair`, or "LSVI-UCB for Fairness" (Algorithm 1) is based on a Least-Squares Value Iteration with an optimistic Upper-Confidence Bound, as in LSVI-UCB Jin et al. (2020). For each $H$-step episode $k$, `L-UCBFair` maintains estimates for $Q_r^k, Q_g^k$ and a time-indexed policy $\pi^k$. In each episode, `L-UCBFair` updates $Q_r^k, Q_g^k$, interacts with the environment, and updates the dual variable $\nu_k$, which is constant over episode $k$.

**LSVI-UCB Jin et al. (2020)**  The estimation of $Q$ is challenging, as it is impossible to iterate over all $s, a$ pairs when $\mathcal{S}$ and $\mathcal{A}$ are continuous and $\mathbf{P}$ is unknown. LSVI parameterizes $Q_h^\star(s, a)$ by the linear form $\mathbf{w}_h^\top \phi(s, a)$, as used by Jin et al. (2020), and updates

$$\mathbf{w}_h \leftarrow \underset{\mathbf{w} \in \mathbb{R}^d}{\operatorname{argmin}} \sum_{\tau=1}^{k-1} \left[ r_h \left( s_h^\tau, a_h^\tau \right) + \max_{a \in \mathcal{A}} Q_{h+1} \left( s_{h+1}^\tau, a \right) - \mathbf{w}^\top \phi \left( s_h^\tau, a_h^\tau \right) \right]^2 + \varsigma \|\mathbf{w}\|^2.$$

In addition, a "bonus term" $\beta \left( \phi^\top \Lambda_h^{-1} \phi \right)^{1/2}$ is added to the estimate of $Q$ in Algorithm 1 to encourage exploration.

**Adaptive Search Policy** Compared to the works of Ding et al. (2021) and Ghosh et al. (2022), the major challenge we face for long-term fairness is a continuous action space $\mathcal{A}$, which renders the

independent computation of $Q_r^k, Q_g^k$ for each action impossible. To handle this issue, we propose an adaptive search policy: Instead of drawing an action directly from a distribution over continuous values, $\pi$ represents a distribution over finitely many ($M$), Voronoi partitions of $\mathcal{A}$. After sampling a region with a softmax scheme, the agent draws action $a$ uniformly at random from it. In addition to defining a Voronoi partitioning of the action space for the adaptive search policy of L-UCBFair (in the supplementary material), we make the following smoothness assumption, which we use to bound the error introduced by this sampling method:

**Assumption 3.2** (Lipschitz). There exists $\rho > 0$, such that $\|\phi(s, a) - \phi(s, a')\|_2 \le \rho \|a - a'\|_2$.

**Dual Update** For L-UCBFair, the update method for estimating $\nu$ in Eq. (4) is also essential. Since $V_{r,1}^\pi(s)$ and $V_{g,1}^\pi(s)$ are unknown, we use $V_{r,1}^k(s)$ and $V_{g,1}^k(s)$ to estimate them. An estimate for $\nu$ is iteratively updated by performing gradient ascent in the Lagrangian (Eq. (4)) with step-size $\eta$, subject to the assumed upper bound $\mathscr{V}$ for $\nu$ (Assumption B.1). A similar method is also used in Ding et al. (2021); Ghosh et al. (2022).

### 3.1.2 Analysis

We next bound the regret and distortion, defined in Eq. (3), realizable by L-UCBFair. We then compare L-UCBFair with existing algorithms for discrete action spaces and discuss the importance of the number of regions $M$ and the maximum distance $\epsilon_I$ from any action to the center of its corresponding Voronoi partition.

**Theorem 3.3** (Boundedness). *With probability $1 - p$, there exists a constant $b$ such that L-UCBFair (Algorithm 1) achieves*

$$\text{Regret}(K) \le \left(b\zeta H^2\sqrt{d^3} + (\mathscr{V} + 1)H\right)\sqrt{K} \quad ; \quad \text{Drtn}(K) \le \frac{b\zeta(1 + \mathscr{V})H^2\sqrt{d^3}}{\mathscr{V}}\sqrt{K}, \quad (5)$$

*for parameter values $\varsigma = 1$ $\epsilon_I \le \frac{1}{2\rho(1 + \mathscr{V})KH\sqrt{d}}$, $\zeta = \log(\log(M)4dHK/p)$, and $\beta = \mathcal{O}(dH\sqrt{\zeta})$.*

For a detailed proof of Theorem 3.3, refer to Appendix B.1. This theorem indicates that L-UCBFair achieves $\mathcal{O}\left(H^2\sqrt{d^3K}\right)$ bounds for both regret and distortion with high probability. Compared to the algorithms introduced by Ding et al. (2021) and Ghosh et al. (2022), which work with discrete action space, L-UCBFair guarantees the same asymptotic bounds on regret and distortion.

### 3.2 R-TD3

Because assumptions in Section 3.1 (e.g., the linear MDP assumption) are often violated in the real world, we also consider deep reinforcement learning methods as a flexible alternative. Concretely, we apply "Twin-Delayed Deep Deterministic Policy Gradient" (TD3) Fujimoto et al. (2018), with an implementation and default parameters provided by the open-source package "Stable Baselines 3" Raffin et al. (2021), on a Lagrangian relaxation of the long-term fairness problem with time-dependent multipliers for the reward and utility terms (Eq. (6)). We term this specific algorithm for long-term fairness R-TD3. While such methods lack provable safety guarantees, they may still confirm our hypothesis that agents trained via RL can learn to sacrifice short-term utility in order to drive dynamics towards preferable long-term states and explicitly incorporate dynamical control objectives provided as functions of state.

To treat the long-term fairness problem (Eq. (1)) using unconstrained optimization techniques (i.e., methods like TD3), we consider a time-dependent Lagrangian relaxation: We train R-TD3 to optimize

$$\min_\pi \mathop{\mathbb{E}}_{a_\tau \sim \pi(s_\tau)} \left[\sum_{\tau=1}^{H} \left((1 - \lambda_\tau)\mathscr{L}(s_\tau, a_\tau) + \lambda_\tau \mathscr{D}(s_\tau, a_\tau)\right)\right], \quad (6)$$

where $s_{\tau+1} \sim \mathbf{P}(s_\tau, a_\tau)$, $\lambda_\tau = \tau/H$.

Strictly applied, myopic fairness constraints can lead to undesirable dynamics and equilibria Raab and Liu (2021). Eq. (6) relaxes these constraints (hard $\rightarrow$ soft) for the near future while emphasizing them long-term. Thus, we hope to develop classifiers that learn to transition to more favorable equilibria.

**Algorithm 1** `L-UCBFair`

---

**Input:** A set of points $\{I_0, I_1, \cdots, I_M\}$ satisfy Definition B.2. $\epsilon_I = \frac{1}{2\rho(1+\chi)KH}$.
$\nu_1 = 0$. $w_{r,h} = w_{g,h} = 0$. $\alpha = \frac{\log(M)K}{2(1+\chi+H)}$. $\eta = \chi/\sqrt{KH^2}$. $\beta = C_1 dH\sqrt{\log(4\log MdT/p)}, \varsigma = 1$.
**for** episode $k = 1, 2, ..., K$ **do**
    Receive the initial state $s_1^k$.
    **for** step $h = H, H-1, \cdots, 1$ **do**
        $\Lambda_h^k \leftarrow \sum_{\tau=1}^{k-1} \phi\left(s_h^\tau, a_h^\tau\right) \phi\left(s_h^\tau, a_h^\tau\right)^T + \varsigma\mathbf{I}$
        **for** $j \in \{r, g\}$ **do**
            $w_{j,h}^k \leftarrow \left(\Lambda_h^k\right)^{-1}\left[\sum_{\tau=1}^{k-1}\phi\left(s_h^\tau, a_h^\tau\right)\left(j\left(s_h^\tau, a_h^\tau\right) + V_{j,h+1}^k\left(s_{h+1}^\tau\right)\right)\right]$
        **end for**
        **for** iteration $i = 1, \cdots, M$ and index $j \in \{r, g\}$ **do**
            $\xi_{i,j} \leftarrow \left(\phi(\cdot, I_i)^T\left(\Lambda_h^k\right)^{-1}\phi(\cdot, I_i)\right)^{1/2}, \quad Q_{j,h}^k(\cdot, I_i) \leftarrow \min\left[\left\langle w_{j,h}^k, \phi(\cdot, I_i)\right\rangle + \beta\xi_{i,j}, H\right]$
        **end for**
        $\text{SM}_{h,k}(I_i \mid \cdot) = \frac{\exp\left(\alpha\left(Q_{r,h}^k(\cdot, I_i) + \nu_k Q_{g,h}^k(\cdot, I_i)\right)\right)}{\sum_j \exp\left(\alpha\left(Q_{r,h}^k(\cdot, I_j) + \nu_k Q_{a,h}^k(\cdot, I_j)\right)\right)}$
        $\pi_h^k(a \mid \cdot) \leftarrow \frac{1}{\int_{b \in \mathcal{I}(a)} db}\text{SM}_{h,k}(I(a) \mid \cdot)$
        $V_{r,h}^k(\cdot) \leftarrow \int_{a \in \mathcal{A}} \pi_h^k(a \mid \cdot)Q_{r,h}^k(\cdot, a)da, \quad V_{g,h}^k(\cdot) \leftarrow \int_{a \in \mathcal{A}} \pi_h^k(a \mid \cdot)Q_{g,h}^k(\cdot, a)da$
    **end for**
    **for** step $h = 1, \cdots, H$ **do**
        Compute $Q_{r,h}^k\left(s_h^k, I_i\right), Q_{g,h}^k\left(s_h^k, I_i\right), \pi\left(I_i \mid s_h^k\right)$.
        Take action $a_h^k \sim \pi_h^k\left(\cdot \mid s_h^k\right)$ and observe $s_{h+1}^k$.
    **end for**
    $\nu_{k+1} = \max\left\{\min\left\{\nu_k + \eta\left(\tilde{c} - V_{g,1}^k(s_1)\right), \mathscr{V}\right\}, 0\right\}$
**end for**

---

### 3.3 Baselines

We compare `L-UCBFair` and `R-TD3` to a greedy agent as a proxy for a myopic status quo in which policy is repeatedly determined by optimizing for immediate utility, without regard for the population dynamics. This standard is known as "Repeated Risk Minimization" (Perdomo et al., 2020; Hu and Zhang, 2022), and we implement it using simple gradient descent on the different classes of ($\lambda$-parameterized) objective functions $f$ we consider (Eq. (7)). Having adopted a notion of fairness that relies on "groups", we presuppose different groups of agents indexed by $g$, and denote group-conditioned loss as $\mathscr{L}_g$. The objectives that are approximately minimized for each baseline are

$$\text{Myopic:} \quad f(\pi) = \underset{a \sim \pi}{\mathrm{E}}\left[\mathscr{L}(s, a)\right]. \tag{7a}$$

$$\text{Myopic-Fair:} \quad f_\lambda(\pi) = \underset{a \sim \pi}{\mathrm{E}}\left[(1-\lambda)\mathscr{L}(s, a) + \lambda\mathscr{D}(s, a)\right], \lambda \in (0, 1). \tag{7b}$$

$$\text{Maxmin:} \quad f(\pi) = \underset{a \sim \pi}{\mathrm{E}}\left[\max_g\left(\mathscr{L}_g(s, a)\right)\right]. \tag{7c}$$

$$\text{Maxmin-Fair:} \quad f_\lambda(\pi) = \underset{a \sim \pi}{\mathrm{E}}\left[(1-\lambda)\max_g\left(\mathscr{L}_g(s, a)\right) + \lambda\mathscr{D}(s, a)\right], \lambda \in (0, 1). \tag{7d}$$

The two "Maxmin" objectives are related to distributionally robust optimization, which has been previously explored in the context of fairness (Hashimoto et al., 2018), while the two "Myopic" objectives are more straight-forward. While our baselines do not guarantee constraint satisfaction, the two objectives labelled "-Fair" are nonetheless "constraint aware" in precisely the same way as a firm that (probabilistically) incurs penalties for violating constraints.

## 4 Simulated Environments

To evaluate the proposed algorithms and baselines, we consider a series of binary ($Y \in \{-1, 1\}$) classification tasks on a population of two groups $\mathcal{G} = \{g_1, g_2\}$ modeled according to evolutionary

game theory (using replicator dynamics, as described in Appendix A.1, in the supplementary material). We consider two families of distributions of real-valued features for the population: One that is purely synthetic, for which $X \sim \mathcal{N}(Y, 1)$, independent of group $G$, and one that is based on logistic regressions to real-world data, described in Appendix A.2 (in the supplementary material). Both families of distributions over $X$ are parameterized by the joint distribution $\Pr(Y, G)$. RL agents are trained on episodes of length $H$ initialized with randomly sampled states.

In order to better handle continuous state space, we make the following assumption, which has been used to simplify similar synthetic environments (Raab and Liu, 2021):

**Assumption 4.1** (Well-behaved feature). For the purely synthetic setting, we require $X$ to be a "well-behaved" real-valued feature within each group. That is, for each group $g$, $\Pr(Y=1 \mid G=g, X=x)$ strictly increases in $x$.

As an intuitive example of Assumption 4.1, if $Y$ represents qualification for a fixed loan and $X$ represents credit-score, we require higher credit scores to imply higher likelihood that an individual is qualified for the loan.

**Theorem 4.2** (Threshold Bayes-optimality). *For each group $g$, when Assumption 4.1 is satisfied, the Bayes-optimal, deterministic binary classifier is a threshold policy described by a feature threshold $a_g$ for group $g$. That is, if $X \geq a_g$ then $\hat{Y} = 1$; otherwise, $\hat{Y} = -1$.*

As a result of Theorem 4.2, we consider our action space to be the space of group-specific thresholds, and denote an individual action as the vector $a := (a_1, a_2)$. Nonetheless, we note that Assumption 4.1 is often violated in practice, as it is in our semi-synthetic setting.

## 5 Experimental Results

Do RL agents learn to seek favorable equilibria against short-term utility? Is Lagrangian relaxation Eq. (1) sufficient to encourage this behavior? We give positive demonstrations for both questions.

### 5.1 Losses and Disparities Considered

Our experiments consider losses $\mathscr{L}$ which combine true-positive and true-negative rates, i.e.,

$$\mathscr{L}(s, a) = 1 - \alpha \mathtt{tp}(s, a) - \beta \mathtt{tn}(s, a) \quad ; \quad r(s, a) = \alpha \mathtt{tp}(s, a) + \beta \mathtt{tn}(s, a), \qquad (8)$$

where $\alpha, \beta \in [0, 1]$; $\mathtt{tp}(s, a) = \Pr_{s,a}(\hat{Y}=1, Y=1)$; and $\mathtt{tn}(s, a) = \Pr_{s,a}(\hat{Y}=-1, Y=-1)$.

For disparity $\mathscr{D}$, we consider functions of the form $\mathscr{D}(s, a) = \frac{1}{2}\|\xi_{s,a}(g_1) - \xi_{s,a}(g_2)\|^2$ that measure violations of demographic parity (DP) (Dwork et al., 2012), equal opportunity (EOp) (Hardt et al., 2016), equalized odds (EO), and qualification rate parity (QR), described in Table 1. We note that QR is inconsequential for the baseline agents, which ignore the mutability of the population state.

### 5.2 Results

Our experiments show that algorithms trained with an RL formulation of long-term fairness can drive a reactive population toward states with higher utility and fairness than myopic policies. In Fig. 1, the baseline policies (i.e., subfigures (a) and (b)), which focus on short-term incentives drive disparate qualification rates and attain lower long-term utility than the RL agents, indicating that short-term utility is *misaligned* with desirable dynamics in this example. In subfigure (b) in particular, the baseline policy increasingly violates the static fairness condition with time, agreeing with previous results by Raab and Liu (2021). Meanwhile, the RL algorithms (subfigures (c) and (d)) learn to drive universally high qualification rates, thus allowing policies that capture higher utility with time (subfigure (e)), Fig. 3.

| Func. | $\xi_{s,a}(g)$ |
|---|---|
| DP | $\Pr_{s,a}(\hat{Y}=1 \mid G=g)$ |
| QR | $\Pr_{s,a}(Y=1 \mid G=g)$ |
| EOp | $\Pr_{s,a}(\hat{Y}=1 \mid Y=1, G=g)$ |
| EO | $\begin{bmatrix} \Pr_{s,a}(\hat{Y}=1 \mid Y=1, G=g) \\ \Pr_{s,a}(\hat{Y}=1 \mid Y=-1, G=g) \end{bmatrix}$ |

Table 1: Disparity Functions, where $\mathscr{D}(s, a) = \frac{1}{2}\|\xi_{s,a}(g_1) - \xi_{s,a}(g_2)\|^2$.

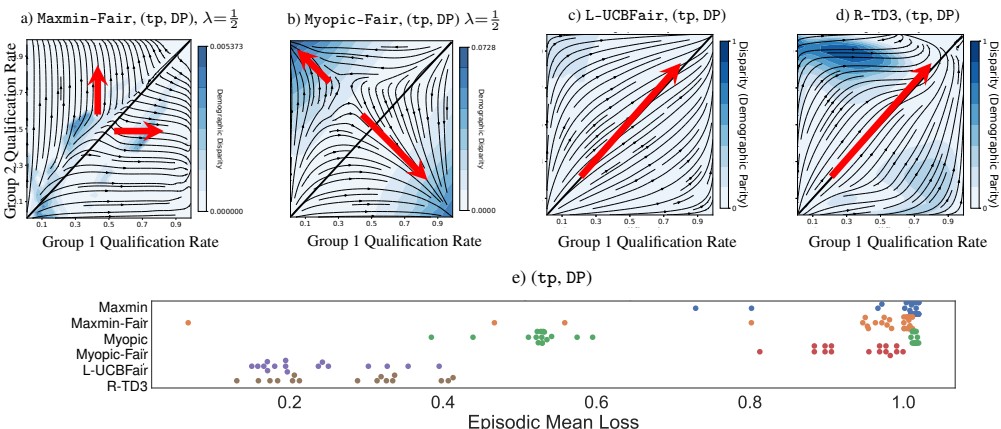

Figure 1: Using a modeled population with scalar features fit to the "Adult" dataset (Dua and Graff, 2017) at each time-step to mirror the evolving qualification rates (Appendix A.2), we compare our baseline algorithms to L-UCBFair and R-TD3 on the same set of 16 initial states with the task of maximizing true positive rates (tp) subject to demographic parity (DP). Plots (a-d) depict evolving group qualification rates under each algorithm with streamlines (*red arrows indicate the author's emphasis*), while shading indicating immediate violations of demographic parity. We remark that the corresponding plots for the non-"Fair" baselines are qualitatively indistinct and omit them for space. In subplot (e), we visualize mean episode loss where $H=150$ for each algorithm.

Our central hypothesis, that long-term fairness via RL may induce an algorithm to sacrifice short-term utility for better long-term outcomes, is clearly demonstrated in the purely synthetic environment depicted by subfigures (a1–a3) of Fig. 2, in which the environment provides higher immediate utility (true-positive rates) but lower long-term utility when a policy chooses initially high acceptance rates. In this case, the RL algorithm (subfigure (a2)) drives the system towards high qualification rates by giving up immediate utility maximized by the myopic agent (subfigure (a1)).

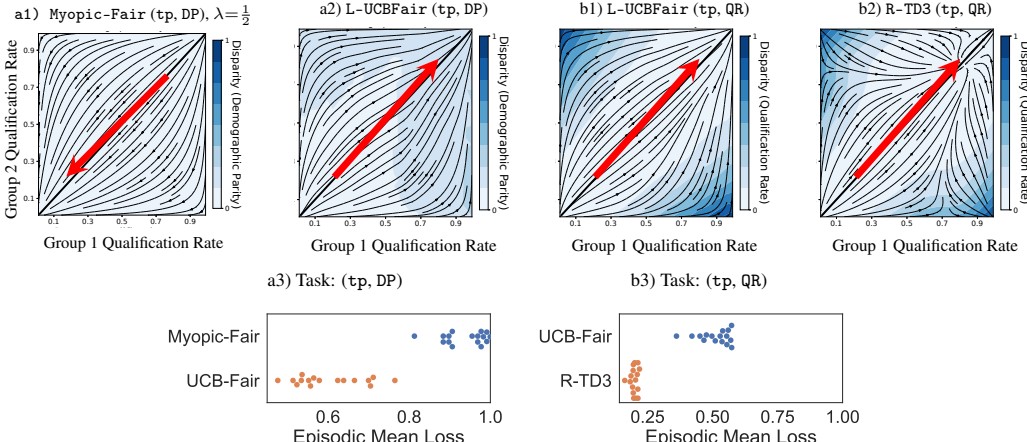

Figure 2: Subfigures (a1–a3) compare a baseline policy to L-UCBFair in a purely synthetic environment on the task maximizing the fraction of true-positive classifications (tp) subject to bounded demographic parity violation (DP). Episode length in this environment is $H = 100$. Subfigures (b1-b3) compare L-UCBFair to R-TD3 on a similar task subject to bounded qualification rate disparity instead. In both environments, modeled agent utilities translate higher acceptance rates to lower qualification rates. Figures (a1, a2) and (b1, b2) depict evolving qualification rates with streamlines (*red arrows indicate the author's emphasis*) and use shading to indicate fairness violations.

With subfigures (b1–b3) of Fig. 2, we demonstrate the capability of RL to incorporate notions of fairness (e.g., qualification rate parity QR), that are impossible to formulate in the myopic setting. In subfigures (b1-b3), both RL agents learn to satisfy qualification rate parity by driving the state of the population towards equal qualification rates by group.

Finally, our experiments also show that RL algorithms without theoretical guarantees may be applicable to long-term fairness. In Fig. 2 subfigure (b2), R-TD3 achieves similar qualitative behavior as L-UCBFair (subfigure (b1)), that is, driving high qualification at the expense of short-term utility) while achieving lower episodic mean loss (subfigure (b3)).

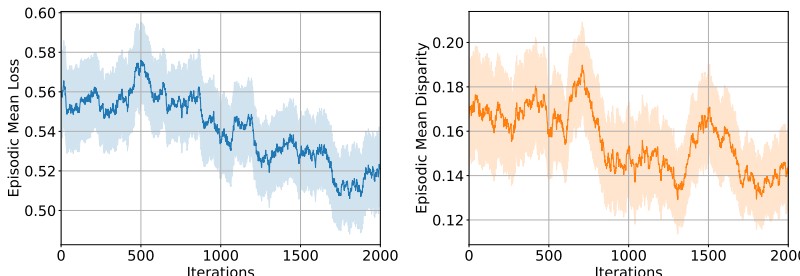

Figure 3: The mean episodic loss (left) and disparity (right) within a 20-step sliding window obtained by L-UCBFair for the (tp, DP) setting of Fig. 1 (a2). We emphasize that both loss and disparity decrease in time, as required for the sublinear regret and distortion guaranteed by (Theorem 3.3).

$$(\mathtt{tp} + 0.8\mathtt{tn}, \mathtt{DP})$$

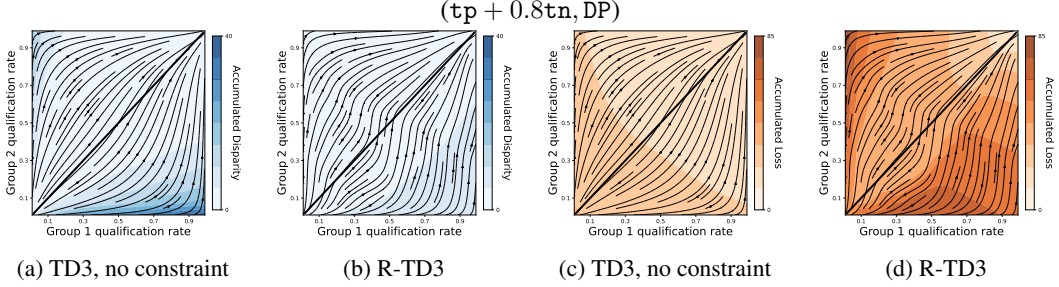

(a) TD3, no constraint      (b) R-TD3      (c) TD3, no constraint      (d) R-TD3

Figure 4: Comparison of total **accumulated** disparity (blue) and loss (orange) over a single episode of $H = 150$ steps as functions of initial state (initial qualification rates), for R-TD3 and TD3 (no constraint; $\lambda = 0$) in the semi-synthetic environment (Adult dataset). This figure indicates that the increasingly weighted fairness term in the objective of R-TD3 can play a role in more rapidly reducing cumulative disparity, in this case by driving the system towards the line of equal qualification rates, at the cost of increased cumulative loss.

**Limitations**    Despite the potential, highlighted by our experiments, for RL-formulation of fairness to drive positive long-term social outcomes, it is not possible to truly validate such approaches without deployment on actual populations, which may be practically and ethically fraught. In addition, violations of fairness or decreased utility may be difficult to justify to affected populations and stakeholders, especially when the bounds provided by L-UCBFair, while as tight as any known, rely on assumptions that may be violated in practice.

**Concluding remarks**    Machine learning techniques are frequently deployed in settings in which affected populations will *react* to the resulting policy, closing a feedback loop that must be accounted for. In such settings, algorithms that prioritize immediate utility or static notions of fairness may yield dynamics that are *misaligned* with these objectives long-term.

In this paper, we have reformulated long-term fairness as an online reinforcement learning problem (Eq. (1)) to address the importance of dynamics. We have shown that algorithmic solutions to this problem (e.g., L-UCBFair) are capable of simultaneous theoretical guarantees regarding cumulative loss and disparity (violations of fairness). We have also shown that these guarantees can be relaxed in practice to accommodate a wider class of RL algorithms, such as R-TD3. Finally, we emphasize again that the RL framework of long-term fairness allows notions of disparity inherent to the *state* of a population to be explicitly treated, while such definitions are inoperable in the standard, myopic framework of fairness. We hope that our contributions spur interest in long-term mechanisms and incentive structures for machine learning to be a driver of positive social change.

**Acknowledgements**    This work is partially supported by the National Science Foundation (NSF) under grants IIS-2143895, IIS-2112471, IIS-2040800, and CCF-2023495.

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

# A  Simulated Environments: Supplementary

## A.1  Replicator Dynamics

Our use of replicator dynamics closely mirrors that of Raab and Liu (2021) as an "equitable" model of a population: Individuals my be modeled identically, independently of group membership, yet persistent outcome disparities may emerge from disparate initial conditions between groups. For our experiments, we parameterize the evolving distribution $\Pr(X, Y \mid G)$, assuming constant group sizes, in terms of "qualification rates" $q_g := \Pr(Y{=}1 \mid G{=}g)$ and update these qualification rates according to the discrete-time replicator dynamics:

$$q_g[t + 1] = q_g[t]\frac{W_1^g[t]}{\overline{W}^g[t]}; \quad \overline{W}^g[t] := W_1^g q_g + (1 - q_g)W_{-1}^g.$$

In this model, the *fitness* $W_y^g > 0$ of label $Y{=}y$ in group $G{=}g$ may be interpreted as the "average utility to the individual" in group $g$ of possessing label $y$, and thus relative *replication rate* of label $y$ in group $g$, as agents update their labels by mimicking the successful strategies of in-group peers. We model $W_y^g$ in terms of acceptance and rejection rates with a group-independent utility matrix $U$:

$$W_y^g = \sum_{\hat{y}\in\{-1,1\}} U_{y,\hat{y}} \Pr\Big(\hat{Y}{=}\hat{y} \mid Y{=}y, G{=}g\Big).$$

We choose the matrix $U$ to eliminate dominant strategies (i.e., when agents universally prefer one label over another, independent of classification), assert that agents always prefer acceptance over rejection, and to imply that the costs of qualification are greater than the costs of non-qualification among accepted individuals. While other parameterizations of $U$ are valid, this choice of parameters guarantees internal equilibrium of the replicator dynamics for a Bayes-optimal classifier and "well-behaved" scalar-valued feature $X$, such that $\Pr(Y{=}1 \mid X{=}x)$ is monotonically increasing in $x$ (Raab and Liu, 2021).

## A.2  Data Synthesis and Processing

In addition to a synthetic distribution, for which we assume $X \sim \mathcal{N}(Y, 1)$, independent of $G$, for all time, we also consider real-world distributions when simulating and comparing algorithms for "long-term fairness". In both cases, as mentioned above, we wish to parameterize distributions in terms of qualification rates $q_g$. As we perform binary classification on discrete groups and scalar-valued features, in addition to parameterizing a distribution in terms of $q_g$, we desire a scalar-valued feature for each example, rather than the multi-dimensional features common to real-world data.

Our solution is to use an additional learning step for "preprocessing": Given a static dataset $\mathcal{D}$ from which $(X', Y, G)$ is drawn i.i.d., (e.g., the "Adult Data Set" Dua and Graff (2017)), where $G$ varies over an individual's sex and $Y$ corresponds to whether an individual has more than 50,000 USD in annual income, at each time-step, we train a stochastic binary classifier $\tilde{a}$, such that $\hat{Y}' \sim \tilde{a}(X', G)$ with a loss that re-weights examples by label value according to $q_g$. That is, at each time step, we solve:

$$\min_{\tilde{a}} \quad \underset{\tilde{a},\mathcal{D}}{\mathrm{E}} \Big[w(X', Y, G)L(Y, \hat{Y}')\Big],$$

where

$$w(X', Y, G) = \left(\frac{1 + Y}{2}\right)\left(\frac{q}{\Pr(Y{=}1 \mid G)}\right) + \left(\frac{1 - Y}{2}\right)\left(\frac{1 - q}{\Pr(Y{=}{-}1 \mid G)}\right),$$

and $L$ is zero-one loss. In our experiments, we choose $\tilde{a}$ according to logistic regression. We interpret $\Pr(\hat{Y}'{=}1)$, drawn from the learned "preprocessing" function $\tilde{a}$, as a new, scalar feature value $X \in \mathbf{R}$ mapped from from higher-dimensional features $X'$. The threshold policy ultimately operates on $X$.

Assumption 4.1 is as hard to satisfy in general as solving the Bayes-optimal binary classification task over higher-dimensional features. Nonetheless, we expect Assumption 4.1 to be approximately satisfied by such a "preprocessing" pipeline. It is important to note that the behavior introduced by retraining a logistic classifier at each time step can yield very different qualitative behavior when compared to the static, purely synthetic distribution, as the distribution of $X$ conditioned on $Y$ and $G$ *also* mutates, and Assumption 4.1 may be violated.

# B    Deferred Proofs

Without loss of generality, we assume $\|\phi(s,a)\| \leq 1$ for all $(s,a) \in \mathcal{S} \times \mathcal{A}$, and $\max\{\|\mu_h(\mathcal{S})\|, \|\theta_h\|\} \leq \sqrt{d}$ for all $h \in [H]$.

**Assumption B.1** ($\mathcal{V}$ bound for $\nu^*$)**.** We define a parameter $\mathcal{V}$ that is assumed to bound the optimal value of $\nu$ in Eq. (4). For $\bar{\pi}$ and $\gamma > 0$ satisfying Slater's Condition (Assumption 2.1),

$$\nu^* \leq \frac{V_{r,1}^{\pi*}(s_1) - V_{r,1}^{\pi}(s_1)}{\gamma} \leq \frac{H}{\gamma} := \mathcal{V}.$$

In Assumption B.1, $\mathcal{V} = H/\gamma$, upper-bounds the optimal dual variable $\nu^*$ as an input to `L-UCBFair`.

**Definition B.2.** Given a set of distinct actions $I = \{I_0, \cdots, I_M\} \subset \mathcal{A}$, where $\mathcal{A}$ is a closed set in Euclidean space, define $\mathcal{I}_i = \{a \colon \|a - I_i\|_2 \leq \|a - I_j\|_2, \forall j \neq i\}$ as the subset of actions closer to $I_i$ than to $I_j$, i.e., the Voronoi region corresponding to locus $I_i$, with tie-breaking imposed by the order of indices $i$. Also define the locus function $I(a) = \min_i \arg\min_{I_i} \|a - I_i\|_2$.

**Lemma B.3.** *The Voronoi partitioning described above satisfies $\mathcal{I}_i \cap \mathcal{I}_j = \varnothing, \forall i \neq j$ and $\cup_{i=1}^M \mathcal{I}_i = \mathcal{A}$.*

*Proof.* We will begin by proving $\mathcal{I}_i \cap \mathcal{I}_j = \varnothing$ for all $i \neq j$. To establish this, we will assume the contrary, that there exist indices $i$ and $j$ such that $\mathcal{I}_i \cap \mathcal{I}_j \neq \varnothing$. Without loss of generality, assume $i > j$. We will denote an arbitrary action within the interaction of $\mathcal{I}_i$ and $\mathcal{I}_j$ as $a' \in \mathcal{A}$.

Considering that $a' \in \mathcal{I}_i$, according to the given Definition B.2, we can infer that $|a' - I_i|_2 < |a' - I_j|_2$ (since $i > j$). However, this assertion contradicts the fact that $a' \in \mathcal{I}_j$, which implies $|a' - I_j|_2 \leq |a' - I_i|_2$. Therefore, $\mathcal{I}_i \cap \mathcal{I}_j = \varnothing$ for all $i \neq j$.

We then proof $\cup_{i=1}^M \mathcal{I}_i = \mathcal{A}$. Since $\mathcal{A}$ is a closed set, for any $a \in \mathcal{A}$, there must be a $i \in \{1, 2, \cdots, M\}$, such that $d_{a,i} = \|a - I_i\|_2 \leq \|a - I_j\|_2 = d_{a,j}, \forall j$. If $d_{a,i} < d_{a,j}$ strictly holds for all $j$, then $a \in \mathcal{I}_i$. Otherwise define a set $\mathcal{J} = \{j | d_{a,j} = d_{a,i}\}$, then $a \in \mathcal{I}_{j'}$ where $j' = \arg\min_{j \in \mathcal{J}} j$. $\qquad\square$

**Theorem B.4.** *If the number $M$ of distinct loci or regions partitioning $\mathcal{A}$ is sufficiently large, there exists a finite set of loci $I$ such that $\forall a \in \mathcal{I}_i, i \in M, \|a - I_i\|_2 \leq \epsilon_I$.*

*Proof.* Since $\mathcal{A}$ is closed in euclidean space, denote $N$ the dimension, $d = \sup_{a,a' \in \mathcal{A}} \|a - a'\|_2$. Define an Orthonormal Basis $\{e_1, e_2, \cdots, e_N\}$. Randomly choose a point $a \in \mathcal{A}$, Set it as $I_0$, then we form a set of loci $I = \{I_0 + \sum_{i=1}^N \epsilon k_i e_i | I_0 + \sum_{i=1}^N \epsilon k_i e_i \in \mathcal{A}, k_i \in \mathcal{Z}, -\lceil \frac{d}{\epsilon} \rceil \leq k_i \leq \lceil \frac{d}{\epsilon} \rceil\}$. We know that $|I| \leq \left(2 \lceil \frac{d}{\epsilon} \rceil\right)^N$. It's not hard to verify that $\|a - I_i\|_2 \leq \frac{\epsilon}{2}\sqrt{2}^{N-1}, \forall a \in \mathcal{I}_i$. Taken $\epsilon_I = \frac{\epsilon}{2}\sqrt{2}^{N-1}$ yields the statement.

$\qquad\square$

## B.1    Proof of Theorem 3.3

**Theorem 3.3** (Boundedness)**.** *With probability $1 - p$, there exists a constant $b$ such that `L-UCBFair` (Algorithm 1) achieves*

$$\text{Regret}(K) \leq \left(b\zeta H^2 \sqrt{d^3} + (\mathcal{V} + 1)H\right)\sqrt{K} \quad; \quad \text{Drtn}(K) \leq \frac{b\zeta(1 + \mathcal{V})H^2\sqrt{d^3}}{\mathcal{V}}\sqrt{K}, \quad (5)$$

*for parameter values $\varsigma = 1$ $\epsilon_I \leq \frac{1}{2\rho(1+\mathcal{V})KH\sqrt{d}}$, $\zeta = \log(\log(M)4dHK/p)$, and $\beta = \mathcal{O}(dH\sqrt{\zeta})$.*

**Outline** The outline of this proof simulates the proof in Ghosh et al. (2022). For brevity, denote $\mathbb{P}_h V_{j,h+1}^{\pi}(s,a) = \mathbb{E}_{s' \sim \mathbb{P}_h(\cdot|s,a)} V_{j,h+1}^{\pi}(s')$ for $j = r, g$. Then

$$Q_{j,h}^{\pi}(s,a) = \left(r_h + \mathbb{P}_h V_{j,h+1}^{\pi}\right)(s,a) \tag{9}$$

$$V_{j,h}^{\pi}(s) = \left\langle \pi_h(\cdot \mid s)Q_{j,h}^{\pi}(s,\cdot)\right\rangle_{\mathcal{A}} \tag{10}$$

$$\left\langle \pi_h(\cdot \mid s), Q_{j,h}^{\pi}(s,\cdot)\right\rangle_{\mathcal{A}} = \sum_{a \in \mathcal{A}} \pi_h(a \mid s)Q_{j,h}^{\pi}(s,a) \tag{11}$$

Similar to Efroni et al. (2020), we establish

$$\text{Regret}(K) + \nu\text{Distortion}(K)$$

$$= \sum_{k=1}^{K}\left(V_{r,1}^{\pi^*}(s_1) - V_{r,1}^{\pi_k}(s_1)\right) + \nu\sum_{k=1}^{K}\left(b - V_{g,1}^{\pi_k}(s_1)\right)$$

$$\leq \underbrace{\sum_{k=1}^{K}\left(V_{r,1}^{\pi^*}(s_1) + \nu_k V_{g,1}^{\pi^*}(s_1)\right) - \left(V_{r,1}^{k}(s_1) + \nu_k V_{g,1}^{k}(s_1)\right)}_{\mathcal{T}_1}$$

$$+ \underbrace{\sum_{k=1}^{K}\left(V_{r,1}^{k}(s_1) - V_{r,1}^{\pi_k}(s_1)\right) + \nu\sum_{k=1}^{K}\left(V_{g,1}^{k}(s_1) - V_{g,1}^{\pi_k}(s_1)\right)}_{\mathcal{T}_2}$$

$$+ \underbrace{\frac{1}{2\eta}\nu^2 + \frac{\eta}{2}H^2 K}_{\mathcal{T}_3}$$

$$(12)$$

$\mathcal{T}_3$ is easily bounded if $\eta$. The major task remains bound $\mathcal{T}_1$ and $\mathcal{T}_2$.

**Bound $\mathcal{T}_1$ and $\mathcal{T}_2$.** We have following two lemmas.

**Lemma B.5** (Boundedness of $\mathcal{T}_1$). *With probability $1 - p/2$, we have $\mathcal{T}_1 \leq KH\left(\frac{\log(M)}{\alpha} + 2(1 + \mathscr{V})H\rho\epsilon_I\sqrt{\frac{dK}{\varsigma}}\right)$. Specifically, if $\alpha = \frac{\log(M)K}{2(1+\mathscr{V}+H)}$ and $\varsigma = 1$, we have $\mathcal{T}_1 \leq 2H(1 + \mathscr{V} + H) + 2KH^2(1 + \mathscr{V})\rho\epsilon_I\sqrt{dK}$ with probability $1 - p/2$.*

**Lemma B.6** (Boundedness of $\mathcal{T}_2$). *Ghosh et al. (2022) With probability $1 - p/2, \mathcal{T}_2 \leq \mathcal{O}\left((\nu + 1)H^2\zeta\sqrt{d^3K}\right)$, where $\zeta = \log[\log(M)4dHK/p]$.*

Lemma B.6 follows the same logic in Ghosh et al. (2022), and we delay the proof of Lemma B.5 to Section B.2. Now we are ready to proof Theorem 3.3.

*Proof.* For any $\nu \in [0, \mathscr{V}]$, with prob. $1 - p$,

$$\text{Regret}(K) + \nu\text{Distortion}(K)$$
$$\leq \mathcal{T}_1 + \mathcal{T}_2 + \mathcal{T}_3$$
$$\leq \frac{1}{2\eta}\nu^2 + \frac{\eta}{2}H^2 K + \frac{HK\log M}{\alpha} + 2KH^2(1 + \mathscr{V})\rho\epsilon_I\sqrt{dK} + \mathcal{O}\left((\nu + 1)H^2\zeta\sqrt{d^3K}\right) \quad (13)$$

Taking $\nu = 0, \eta = \frac{\mathscr{V}}{\sqrt{KH^2}}, \alpha = \frac{K\log M}{2(1+\mathscr{V}+H)}, \epsilon_I = \frac{1}{2\rho(1+\mathscr{V})KH\sqrt{d}}$, there exist constant $b$,

$$\text{Regret}(K) \leq \frac{\mathscr{V}H}{2}\sqrt{K} + 2H(1 + \mathscr{V} + H) + 2H^2 K(1 + \mathscr{V})\rho\epsilon_I\sqrt{dK} + \mathcal{O}\left(H^2\zeta\sqrt{d^3K}\right)$$
$$\leq \left(b\zeta H^2\sqrt{d^3} + (\mathscr{V} + 1)H\right)\sqrt{K} = \tilde{\mathcal{O}}(H^2\sqrt{d^3K}).$$

Taking $\nu = \mathscr{V}, \eta = \frac{\mathscr{V}}{\sqrt{KH^2}}, \alpha = \frac{K\log M}{2(1+\mathscr{V}+H)}, \epsilon_I = \frac{1}{2\rho(1+\mathscr{V})KH\sqrt{d}}$,

$$\text{Regret}(K) + \mathscr{V}\text{Distortion}(K) \leq (\mathscr{V} + 1)H\sqrt{K} + (1 + \mathscr{V})\mathcal{O}\left(H^2\zeta\sqrt{d^3K}\right)$$

Following the idea of Efroni et al. (2020), there exists a policy $\pi'$ such that $V_{r,1}^{\pi'} = \frac{1}{K}\sum_{k=1}^{K}V_{r,1}^{\pi_k}, V_{g,1}^{\pi'} = \frac{1}{K}\sum_{k=1}^{K}V_{g,1}^{\pi_k}$. By the occupancy measure, $V_{r,1}^{\pi}$ and $V_{g,1}^{\pi}$ are linear in occupancy measure induced by $\pi$. Thus, the average of $K$ occupancy measure also produces an occupancy measure which induces policy $\pi'$ and $V_{r,1}^{\pi'}$, and $V_{g,1}^{\pi'}$. We take $\nu = 0$ when $\sum_{k=1}^{K}\left(b - V_{g,1}^{\pi_k}(s_1^k)\right) < 0$,

otherwise $\nu = \mathscr{V}$. Hence, we have

$$V_{r,1}^{\pi^*}(s_1) - \frac{1}{K}\sum_{k=1}^{K} V_{r,1}^{\pi_k}(s_1) + \mathscr{V}\max\left((c - \frac{1}{K}\sum_{k=1}^{K} V_{g,1}^{\pi_k}(s_1), 0\right)$$

$$= V_{r,1}^{\pi^*}(s_1) - V_{r,1}^{\pi'}(s_1) + \mathscr{V}\max\left(c - V_{g,1}^{\pi'}(s_1), 0\right)$$

$$\leq \frac{\mathscr{V}+1}{K} H\sqrt{K} + \frac{\mathscr{V}+1}{K}\mathcal{O}\left(H^2\zeta\sqrt{d^3 K}\right) \qquad (14)$$

Since $\mathscr{V} = 2H/\gamma$, and using the result of Lemma B.15, we have

$$\max\left(c - \frac{1}{K}\sum_{k=1}^{K} V_{g,1}^{\pi_k}(s_1^k), 0\right) \leq \frac{\mathscr{V}+1}{K\mathscr{V}}\mathcal{O}\left(H^2\zeta\sqrt{d^3 K}\right)$$

In this section we proof Lemma B.5 and Lemma B.6.

## B.2 Prepare for Lemma B.5

In order to bound $\mathcal{T}_1$ and $\mathcal{T}_2$, we introduce the following lemma.

**Lemma B.7.** *There exists a constant $B_2$ such that for any fixed $p \in (0,1)$, with probability at least $1 - p/2$, the following event holds*

$$\|\sum_{\tau=1}^{k-1}\phi_{j,h}^{\tau}\left[V_{j,h+1}^{k}(s_{h+1}^{\tau}) - \mathbb{P}_h V_{j,h+1}^{k}(s_h^{\tau}, a_h^{\tau})\right]\|_{(\varsigma_h^k)^{-1}} \leq B_2 dHq$$

*for $j \in \{r, g\}$, where $q = \sqrt{\log\left[4(B_1 + 1)\log(M)dT/p\right]}$ for some constant $B_1$.*

We delay the proof of Lemma B.7 to Appendix B.4.

Lemma B.7 shows the bound of estimated value function $V_{j,h}^k$ and value function $V_{j,h}^\pi$ corresponding in a given policy at k. We now introcuce the following lemma appeared in Ghosh et al. (2022). This lemma bounds the difference between the value function without bonus in L-UCBFair and the true value function of any policy $\pi$. This is bounded using their expected difference at next step, plus a error term.

**Lemma B.8.** *Ghosh et al. (2022) There exists an absolute constant $\beta = C_1 dH\sqrt{\zeta}, \zeta = \log(\log(M)4dT/p)$, and for any fixed policy $\pi$, for the event defined in Lemma B.7, we have*

$$\langle\phi(s,a), w_{j,h}^k\rangle - Q_{j,h}^\pi(s,a) = \mathbb{P}_h\left(V_{j,h+1}^k - V_{j,h+1}^\pi\right)(s,a) + \Delta_h^k(s,a)$$

*for some $\Delta_h^k(s,a)$ that satisfies $\left|\Delta_h^k(s,a)\right| \leq \beta\sqrt{\phi(s,a)^T\left(\Lambda_h^k\right)^{-1}\phi(s,a)}$.*

**Lemma B.9.** *Ghosh et al. (2022) With probability at least $1 - p/2$, (for the event defined in Lemma B.7)*

$$Q_{r,h}^\pi(s,a) + \nu_k Q_{g,h}^\pi(s,a) \leq Q_{r,h}^k(s,a) + \nu_k Q_{g,h}^k(s,a) - \mathbb{P}_h\left(V_{h+1}^k - V_{h+1}^{\pi,\nu_k}\right)(s,a)$$

We also introduce the following lemma. This lemma bound the value function by taking L-UCBFair policy and greedy policy.

**Lemma B.10.** *Define $\bar{V}_h^k(\cdot) = \max_a\left[Q_{r,h}^k(\cdot,a) + \nu_k Q_{g,h}^k(\cdot,a)\right]$ the value function corresponding to greedy policy, we have*

$$\bar{V}_h^k(s) - V_h^k(s) \leq \frac{\log M}{\alpha} + 2(1 + \mathscr{V})H\rho\epsilon_I\sqrt{\frac{dk}{\varsigma}}. \qquad (15)$$

*Proof.* Define $a_g$ the solution of greedy policy,

$$\bar{V}_h^k(s) - V_h^k(s) = \left[Q_{r,h}^k(s, a_g) + \nu_k Q_{g,h}^k(s, a_g)\right] \tag{16}$$

$$- \int_a \pi_{h,k}(a \mid s) \left[Q_{r,h}^k(s, a) + \nu_k Q_{g,h}^k(s, a)\right] da \tag{17}$$

$$\leq \left[Q_{r,h}^k(s, a_g) + \nu_k Q_{g,h}^k(s, a_g)\right] \tag{18}$$

$$- \sum_i \mathrm{SM}_\alpha(I_i \mid x) \left[Q_{r,h}^k(x, I_i) + \nu_k Q_{g,h}^k(x, I_i)\right] + 2(1 + \mathcal{V})H\rho\epsilon_I\sqrt{\frac{dk}{\varsigma}} \tag{19}$$

$$\leq \left(\frac{\log\left(\sum_a \exp\left(\alpha\left(Q_{r,h}^k(s, I_i) + \nu_k Q_{g,h}^k(s, I_i)\right)\right)\right)}{\alpha}\right) \tag{20}$$

$$- \sum_i \mathrm{SM}_\alpha(I_i \mid s) \left[Q_{r,h}^k(s, I_i) + \nu_k Q_{g,h}^k(s, I_i)\right] + 2(1 + \mathcal{V})H\rho\epsilon_I\sqrt{\frac{dk}{\varsigma}} \tag{21}$$

$$\leq \frac{\log(M)}{\alpha} + 2(1 + \mathcal{V})H\rho\epsilon_I\sqrt{\frac{dk}{\varsigma}}. \tag{22}$$

The first inequality follows from Lemma B.14 and the second inequality holds because of Proposition 1 in Pan et al. (2019).

## B.3 Proof of Lemma B.5

Now we're ready to proof Lemma B.5.

*Proof.* This proof simulates Lemma 3 in Ghosh et al. (2022).

We use induction to proof this lemma. At step $H$, we have $Q_{j,H+1}^k = 0 = Q_{j,H+1}^\pi$ by definition. Under the event in Lemma B.13 and using Lemma B.8, we have for $j = r, g$,

$$\left|\langle\phi(s, a), w_{j,H}^k(s, a)\rangle - Q_{j,H}^\pi(s, a)\right| \leq \beta\sqrt{\phi(s, a)^T\left(\Lambda_H^k\right)^{-1}\phi(s, a)}$$

Thus $Q_{j,H}^\pi(s, a) \leq \min\left\{\langle\phi(s, a), w_{j,H}^k\rangle + \beta\sqrt{\phi(s, a)^T\left(\Lambda_H^k\right)^{-1}\phi(s, a)}, H\right\} = Q_{j,H}^k(s, a)$.

From the definition of $\bar{V}_h^k$,

$$\bar{V}_H^k(s) = \max_a\left[Q_{r,H}^k(s, a) + \nu_k Q_{g,h}^k(s, a)\right] \geq \sum_a \pi(a \mid x)\left[Q_{r,H}^\pi(s, a) + \nu_k Q_{g,H}^\pi(s, a)\right] = V_H^{\pi,\nu_k}(s)$$

for any policy $\pi$. Thus, it also holds for $\pi^*$, the optimal policy. Using Lemma B.10 we can get

$$V_H^{\pi^*,\nu_k}(s) - V_H^k(s) \leq \frac{\log M}{\alpha} + 2(1 + \mathcal{V})H\rho\epsilon_I\sqrt{\frac{dk}{\varsigma}}$$

Now, suppose that it is true till the step $h + 1$ and consider the step $h$. Since, it is true till step $h + 1$, thus, for any policy $\pi$,

$$\mathbb{P}_h\left(V_{h+1}^{\pi,\nu_k} - V_{h+1}^k\right)(s, a) \leq (H - h)\left(\frac{\log M}{\alpha} + 2(1 + \mathcal{V})H\rho\epsilon_I\sqrt{\frac{dk}{\varsigma}}\right)$$

From (27) in Lemma 10 and the above result, we have for any $(s, a)$

$$Q_{r,h}^\pi(s, a) + \nu_k Q_{g,h}^\pi(s, a) \leq Q_{r,h}^k(s, a) + \nu_k Q_{g,h}^k(s, a) + (H - h)\left(\frac{\log M}{\alpha} + 2(1 + \mathcal{V})H\rho\epsilon_I\sqrt{\frac{dk}{\varsigma}}\right)$$

Hence,

$$V_h^{\pi,\nu_k}(s) \leq \bar{V}_h^k(s) + (H - h)\left(\frac{\log M}{\alpha} + 2(1 + \mathcal{V})H\rho\epsilon_I\sqrt{\frac{dk}{\varsigma}}\right)$$

Now, again from Lemma 11, we have $\bar{V}_h^k(s) - V_h^k(s) \leq \frac{\log(|\mathcal{A}|)}{\alpha}$. Thus,

$$V_h^{\pi,\nu_k}(s) - V_h^k(s) \leq (H - h + 1)\left(\frac{\log M}{\alpha} + 2(1 + \mathcal{V})H\rho\epsilon_I\sqrt{\frac{dk}{\varsigma}}\right)$$

Now, since it is true for any policy $\pi$, it will be true for $\pi^*$. From the definition of $V^{\pi,\nu_k}$, we have

$$\left(V_{r,h}^{\pi^*}(s) + \nu_k V_{g,h}^{\pi^*}(s)\right) - \left(V_{r,h}^k(s) + \nu_k V_{g,h}^k(s)\right) \leq (H - h + 1)\left(\frac{\log M}{\alpha} + 2(1 + \mathcal{V})H\rho\epsilon_I\sqrt{\frac{dk}{\varsigma}}\right)$$

Hence, the result follows by summing over $K$ and considering $h = 1$.

## B.4 Proof of Lemma B.7

We first define some useful sets. Let $\mathcal{Q}_j = \left\{Q \mid Q(\cdot, a) = \min\left\{w_j^T\phi(\cdot, a) + \beta\sqrt{\phi^T(\cdot, a)^T\Lambda^{-1}\phi(\cdot, a)}, H\right\}, a \in \mathcal{A}\right\}$ be the set of Q functions, where $j \in \{r, g\}$. Since the minimum eigen value of $\Lambda$ is no smaller than one so the Frobenius norm of $\Lambda^{-1}$ is bounded.

Let $\mathcal{V}_j = \left\{V_j \mid V_j(\cdot) = \int_a \pi(a \mid \cdot)Q_j(\cdot, a)da; Q_r \in \mathcal{Q}_r, Q_g \in \mathcal{Q}_g, \nu \in [0, \mathcal{V}]\right\}$ be the set of Q functions, where $j \in \{r, g\}$. Define

$$\Pi = \left\{\pi \mid \forall a \in \mathcal{A}, \pi(a \mid \cdot) = \frac{1}{\int_{b \in \mathcal{I}(a)} db}\text{SM}_\alpha\left(Q_r(\cdot, I(a)) + \nu Q_g(\cdot, I(a))\right), \ Q_r \in \mathcal{Q}_r, Q_g \in \mathcal{Q}_g, \nu \in [0, \mathcal{V}]\right\}$$

the set of policies.

It's easy to verify $V_j^k \in \mathcal{V}_j$.

Then we introduce the proof of Lemma B.7. To proof Lemma B.7, we need the $\epsilon$-covering number for the set of value functions(Lemma B.13Ghosh et al. (2022)). To achieve this, we need to show if two $Q$ functions and the dual variable $\nu$ are close, then the bound of policy and value function can be derived(Lemma B.11, Lemma B.12). Though the proof of Lemma B.11 and Lemma B.12 are different from Ghosh et al. (2022), we show the results remain the same, thus Lemma B.13 still holds. We'll only introduce Lemma B.13 and omit the proof.

We now proof Lemma B.11.

**Lemma B.11.** *Let $\pi$ be the policy of* L-UCBFair *corresponding to $Q_r^k + \nu_k Q_g^k$, i.e.,*

$$\pi(a \mid \cdot) = \frac{1}{\int_{b \in \mathcal{I}(a)} db}\text{SM}_\alpha\left(Q_r(\cdot, I(a)) + \nu Q_g(\cdot, I(a))\right) \tag{23}$$

*and*

$$\tilde{\pi}(a \mid \cdot) = \frac{1}{\int_{b \in \mathcal{I}(a)} db}\text{SM}_\alpha\left(\tilde{Q}_r(\cdot, I(a)) + \tilde{\nu}\tilde{Q}_g(\cdot, I(a))\right), \tag{24}$$

*if $\left|Q_j - \tilde{Q}_j\right| \leq \epsilon'$ and $|\nu - \tilde{\nu}| \leq \epsilon'$, then $\left|\int_a (\pi(a \mid x) - \tilde{\pi}(a \mid x))\, da\right| \leq 2\alpha\epsilon'(1 + \mathcal{V} + H)$.*

*Proof.*

$$\left|\int_a (\pi(a \mid x) - \tilde{\pi}(a \mid x))\, da\right| \tag{25}$$

$$= \left|\sum_{i=1}^{M} \int_{a \in \mathcal{I}_i} (\pi(I(a) \mid x) - \tilde{\pi}(I(a) \mid x))\, da\right|$$

$$= \left|\sum_{i=1}^{M} \int_{b \in \mathcal{I}_i} db\, (\pi(I_i \mid x) - \tilde{\pi}(I_i \mid x))\right|$$

$$\leq \sum_{i=1}^{M} \left|\text{SM}_\alpha\left(Q_r(s, I_i) + \nu Q_g(s, I_i)\right) - \text{SM}_\alpha\left(\tilde{Q}_r(s, I_i) + \tilde{\nu}\tilde{Q}_g(s, I_i)\right)\right|$$

$$\leq 2\alpha\left|Q_r(\cdot, I(a)) + \nu Q_g(\cdot, I(a)) - \tilde{Q}_r(\cdot, I(a)) - \tilde{\nu}\tilde{Q}_g(\cdot, I(a))\right| \tag{26}$$

The last inequality holds because of Theorem 4.4 in Epasto et al. (2020). Using Corollary B.17, we have

$$\left| \int_a \left( \pi(a \mid x) - \tilde{\pi}(a \mid x) \right) da \right| \leq 2\alpha\epsilon'(1 + \mathscr{V} + H) \tag{27}$$

Now since we have Lemma B.11, we can further bound the value functions.

**Lemma B.12.** *If* $\left| \tilde{Q}_j - Q_j^k \right| \leq \epsilon'$, *where* $\tilde{Q}_j \in \mathcal{Q}_j$, *then there exists* $\tilde{V}_j \in \mathcal{V}_j$ *such that*

$$\left| V_j^k - \widetilde{V}_j \right| \leq H2\alpha\epsilon'(1 + \mathscr{V} + H) + \epsilon',$$

*Proof.* For any $x$,

$$V_j^k(s) - \widetilde{V}_j(s)$$

$$= \left| \int_a \pi(a \mid s) Q_j^k(s,a) da - \int_a \tilde{\pi}(a \mid s) \tilde{Q}_j(s,a) da \right|$$

$$= \left| \int_a \pi(a \mid s) Q_j^k(s,a) da - \int_a \pi(a \mid s) \tilde{Q}_j(s,a) da + \int_a \pi(a \mid s) \tilde{Q}_j(s,a) da - \int_a \tilde{\pi}(a \mid s) \tilde{Q}_j(s,a) da \right|$$

$$\leq \left| \int_a \pi(a \mid s) \left( Q_j^k(s,a) - \tilde{Q}_j(s,a) \right) da \right| + \left| \int_a \pi(a \mid s) \tilde{Q}_j(s,a) da - \int_a \tilde{\pi}(a \mid s) \tilde{Q}_j(s,a) da \right|$$

$$\leq \epsilon' + H \left| \int_a \left( \pi(a \mid s) - \tilde{\pi}(a \mid s) \right) da \right|$$

$$\leq \epsilon' + H2\alpha\epsilon'(1 + \mathscr{V} + H)$$

Using Lemmas above, we can have the same result presented in Lemma 13 of Ghosh et al. (2022) as following.

**Lemma B.13.** *Ghosh et al. (2022) There exists a* $\tilde{V}_j \in \mathcal{V}_j$ *parameterized by* $\left( \tilde{w}_r, \tilde{w}_g, \tilde{\beta}, \Lambda, \tilde{\mathscr{V}} \right)$ *such that* $\text{dist}\left( V_j, \tilde{V}_j \right) \leq \epsilon$ *where*

$$\left| V_j - \tilde{V}_j \right| = \sup_x \left| V_j(s) - \tilde{V}_r(s) \right|.$$

*Let* $N_\epsilon^{V_j}$ *be the* $\epsilon$-*covering number for the set* $\mathcal{V}_j$, *then,*

$$\log N_\epsilon^{V_j} \leq d \log \left( 1 + 8H \frac{\sqrt{dk}}{\sqrt{\varsigma}\epsilon'} \right) + d^2 \log \left[ 1 + 8d^{1/2}\beta^2 / \left( \varsigma \left( \epsilon' \right)^2 \right) \right] + \log \left( 1 + \frac{\mathscr{V}}{\epsilon'} \right)$$

*where* $\epsilon' = \frac{\epsilon}{H2\alpha(1 + \mathscr{V} + H) + 1}$

**Lemma B.14.** $|Q_{j,h}^k(s,a) - Q_{j,h}^k(s, I(a))| \leq 2H\rho\epsilon_I \sqrt{\frac{dK}{\varsigma}}$.

*Proof.*

$$\left| Q_{j,h}^k(s,a) - Q_{j,h}^k(s, I(a)) \right| \tag{28}$$

$$= \left| w_{j,h}^k(s,a)^T \left( \phi(s,a) - \phi(s, I(a)) \right) \right| \tag{29}$$

$$\leq \| w_{j,h}^k(s,a) \|_2 \| \phi(s,a) - \phi(s, I(a)) \|_2 \tag{30}$$

$$\tag{31}$$

From Lemma B.16 and Assumption 3.2 we get the result.

### B.5 Prior Results

**Lemma B.15.** *Ding et al. (2021) Let $\nu^*$ be the optimal dual variable, and $C \geq 2\nu^*$, then, if*

$$V_{r,1}^{\pi^*}(s_1) - V_{r,1}^{\pi}(s_1) + C\left[c - V_{g,1}^{\pi}(s_1)\right]_+ \leq \delta,$$

*we have*

$$\left[c - V_{g,1}^{\tilde{\pi}}(x_1)\right]_+ \leq \frac{2\delta}{C}.$$

**Lemma B.16.** *Jin et al. (2020) For any $(k,h)$, the weight $w_{j,h}^k$ satisfies*

$$\left\|w_{j,h}^k\right\| \leq 2H\sqrt{dk/\varsigma}$$

**Corollary B.17.** *If $\mathrm{dist}\left(Q_r, \tilde{Q}_r\right) \leq \epsilon'$, $\mathrm{dist}\left(Q_g, \tilde{Q}_g\right) \leq \epsilon'$, and $|\tilde{\nu}_k - \nu_k| \leq \epsilon'$, then, $\mathrm{dist}\left(Q_r^k + \nu_k Q_g^k, \tilde{Q}_r + \tilde{\nu}_k \tilde{Q}_g\right) \leq \epsilon'(1 + \mathscr{V} + H)$.*

# C  Additional Figures

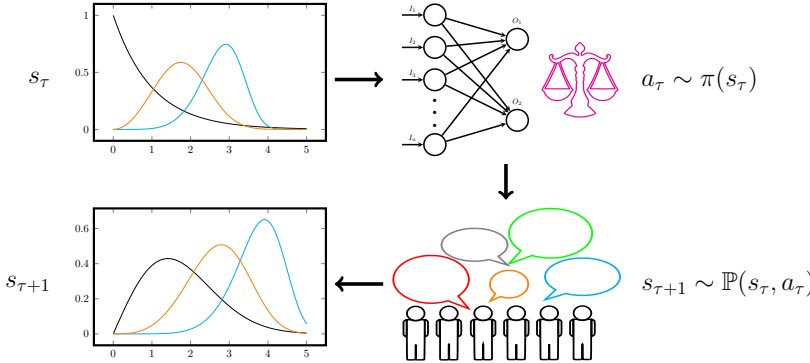

Figure 5: The interaction of an algorithmic classifier and a reactive population. Given state $s_\tau$, the classifier uses policy $\pi$ to select action $a_\tau$. The population, in state $s_\tau$, reacts to $a_\tau$, transitioning state to $s_{\tau+1}$, then the process repeats.

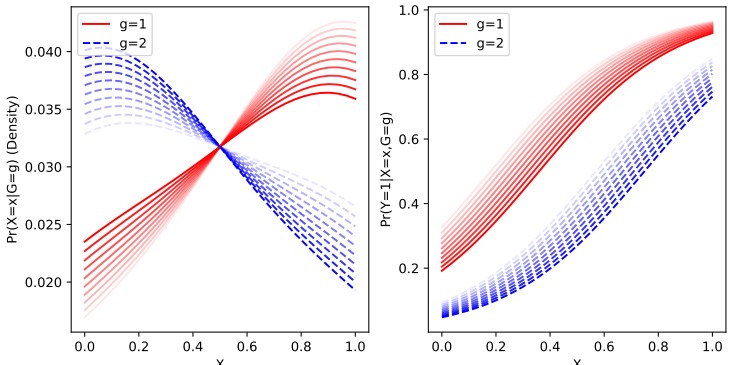

Figure 6: A synthetic distribution is updated according to a dynamical kernel $\mathbf{P}$ based on evolutionary dynamics (Appendix A.1), when a classifier repeatedly predicts $\hat{Y}=1$ iff $X \geq 0.5$. We visualize how the distribution of $X$ and conditional qualification rates $\Pr(Y=1 \mid X)$ change in each group $g \in \{1 \text{ (red, solid)}, 2 \text{ (blue, dashed)}\}$, fading the plotted lines over 10 time steps. In this example, the feature values $X$ in each group decrease with time, while the qualification rates of agents at any fixed value of $X$ decrease.

# D Experiments

## D.1 Experiment Details

**Device and Packages.** We run all the experiment on a single 1080Ti GPU. We implement the `R-TD3` agent using StableBaseline3 (Raffin et al., 2021). The neural network is implemented using Pytorch (Paszke et al., 2019). Figures were generated using code included in the supplementary material in less than 24 hours on a single Nvidia RTX A5000.

**Using a Neural Network to Learn $\phi$.** We use a multi-layer perceptron to learn $\phi$. Specifically, we sample 100000 data points using a random policy, storing $s$, $a$, $r$ and $g$. The inputs of the network are state and action, passing through fully connected (fc) layers with size 256, 128, 64, 64. ReLU is used as activation function between fc layers, while a SoftMax layer is applied after the last fc layer. We treat the outcome of this network as $\phi$. To learn $\phi$, we apply two separated fc layers (without bias) with size 1 to $\hat{\phi}$ and treat the outputs as predicted $r$ and predicted $g$. A combination of MSE losses of $r$ and $g$ are adopted. We use Adam as the optimizer. Weight decay is set to 1e-4 and learning rate is set to 1e-3, while batch size is 128.

Note that, $\hat{\phi}$ is linear regarding $r$ and $g$, but the linearity of transition kernel cannot be captured using such a schema. Therefore, equivalently we made an assumption that there always exists measure $\mu_h$ such that for given $\hat{\phi}$, the linearity of transition kernel holds. It's a stronger assumption than Assumption 3.1.

## D.2  Maximize True-Positives; Synthetic Data

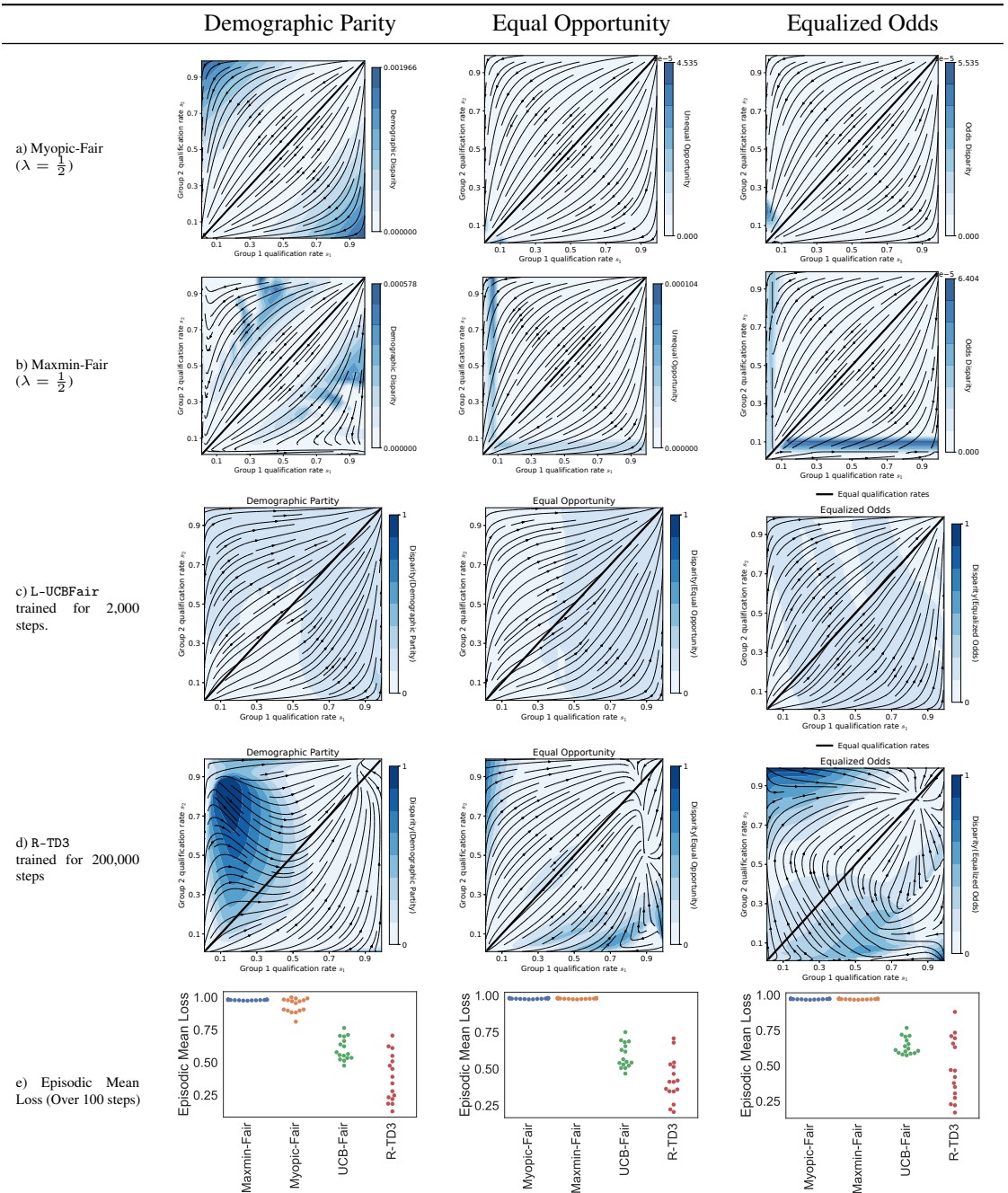

Table 2: A comparison of baseline and RL policies in a purely synthetic environment with the objective of maximizing the cumulative true positive fraction of a binary classification task (Section 4), subject to three fairness constraints (Section 5.1) (columns). In all cases, the baseline, greedy policies drive the system (indicated by streamlines) to promote low qualification rates in each group (lower left corner of each plot), while the RL agents are able to drive the system to more favorable equilibria characterized by higher qualification rates (upper right). Shading represents local (noncumulative) violations of fairness.

## D.3 Maximize True-Positives; "Adult" Data

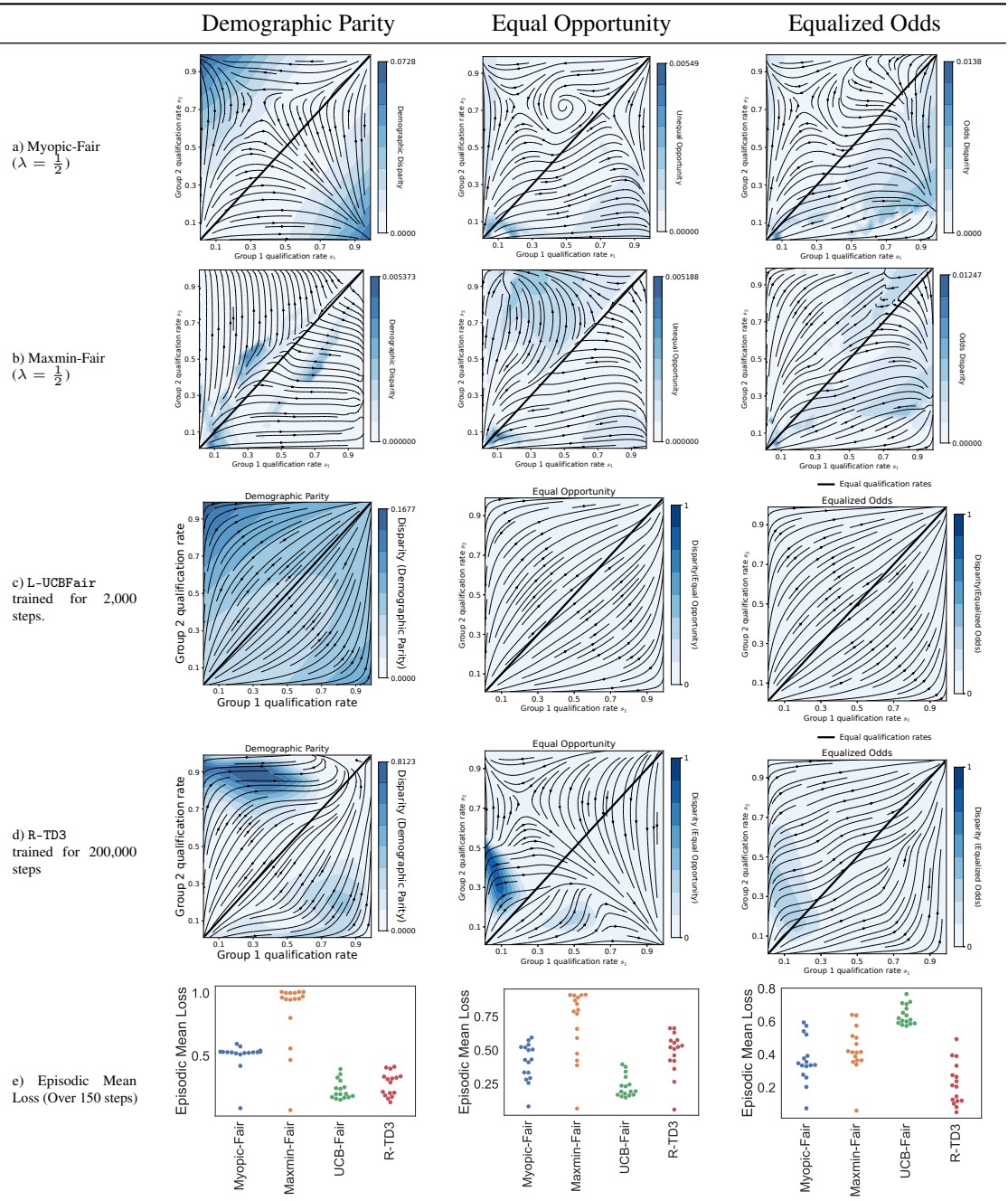

Table 3: A comparison of baseline and RL policies in a semi-synthetic environment incorporating data synthesized from the UCI "Adult Data Set" (Dua and Graff, 2017), as detailed in section Appendix A.2. Policies share the objective of maximizing the cumulative true positive fraction of a binary classifications (Section 4), subject to three fairness constraints (Section 5.1) (columns). The qualitative difference between the synthetic environment (Table 2) and this, modified environment can be largely explained by the fact that $\Pr(Y=1|X=x))$ is not actually monotonically increasing in $x$, as stipulated by Assumption 4.1. (Appendix A.2)

**D.4 Maximize True-Positives** $+0.8$ **True-Negatives; Synthetic Data**

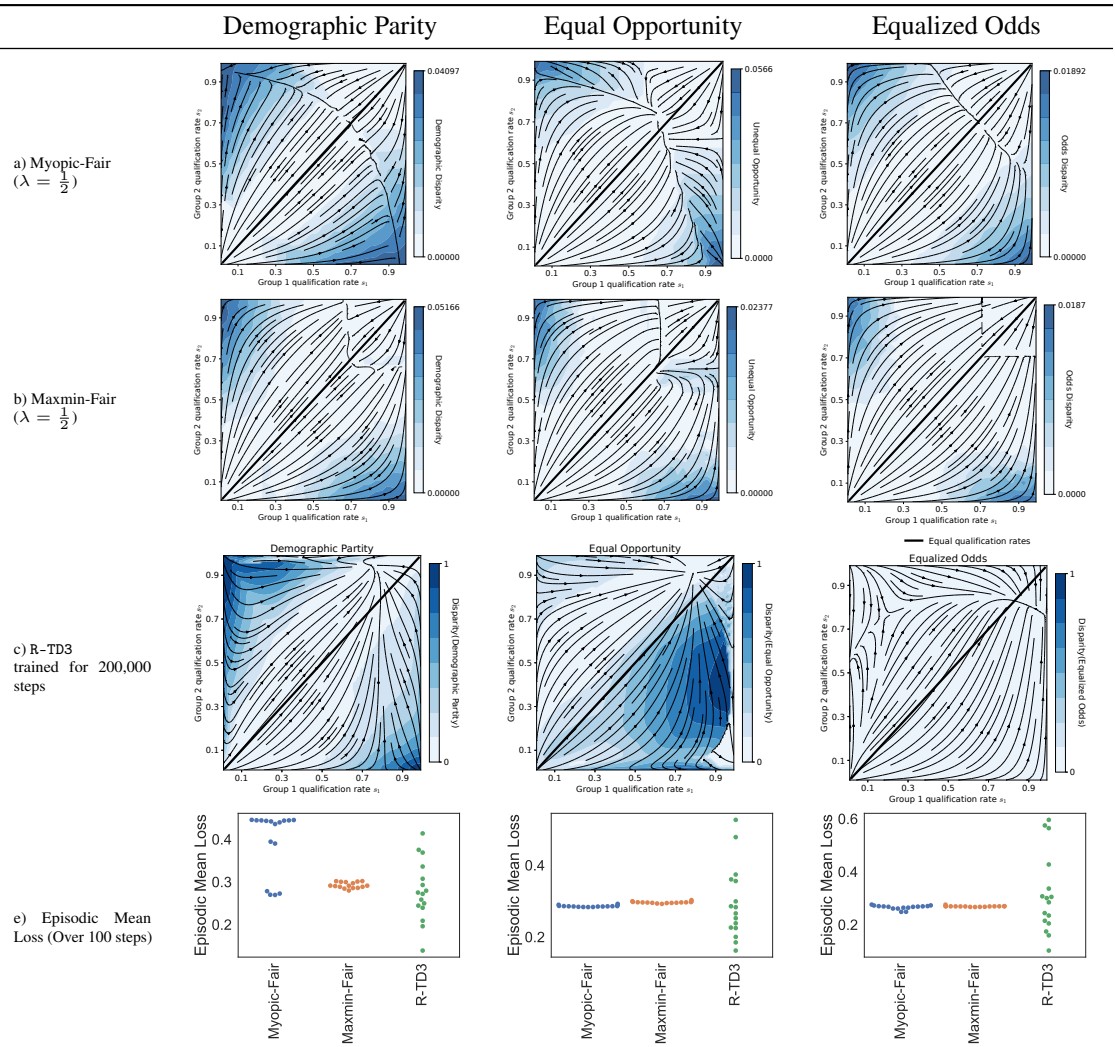

Table 4: A comparison of baseline and R-TD3 policies in a purely synthetic environment with the objective of maximizing a weighted combination of the cumulative true positive and true negative fractions of a binary classification task (Section 4), subject to three fairness constraints (Section 5.1) (columns). In all cases, the baseline, greedy policies drive the system (indicated by streamlines) to a 1-dimensional manifold in the space of group qualification rates that is bounded away from universal qualification (upper right corner of each plot), while the RL agents are generally able to drive the system closer to this favorable equilibrium. Shading represents local violations of fairness.

## D.5   Maximize True-Positives $+0.8$ True-Negatives; "Adult" Data

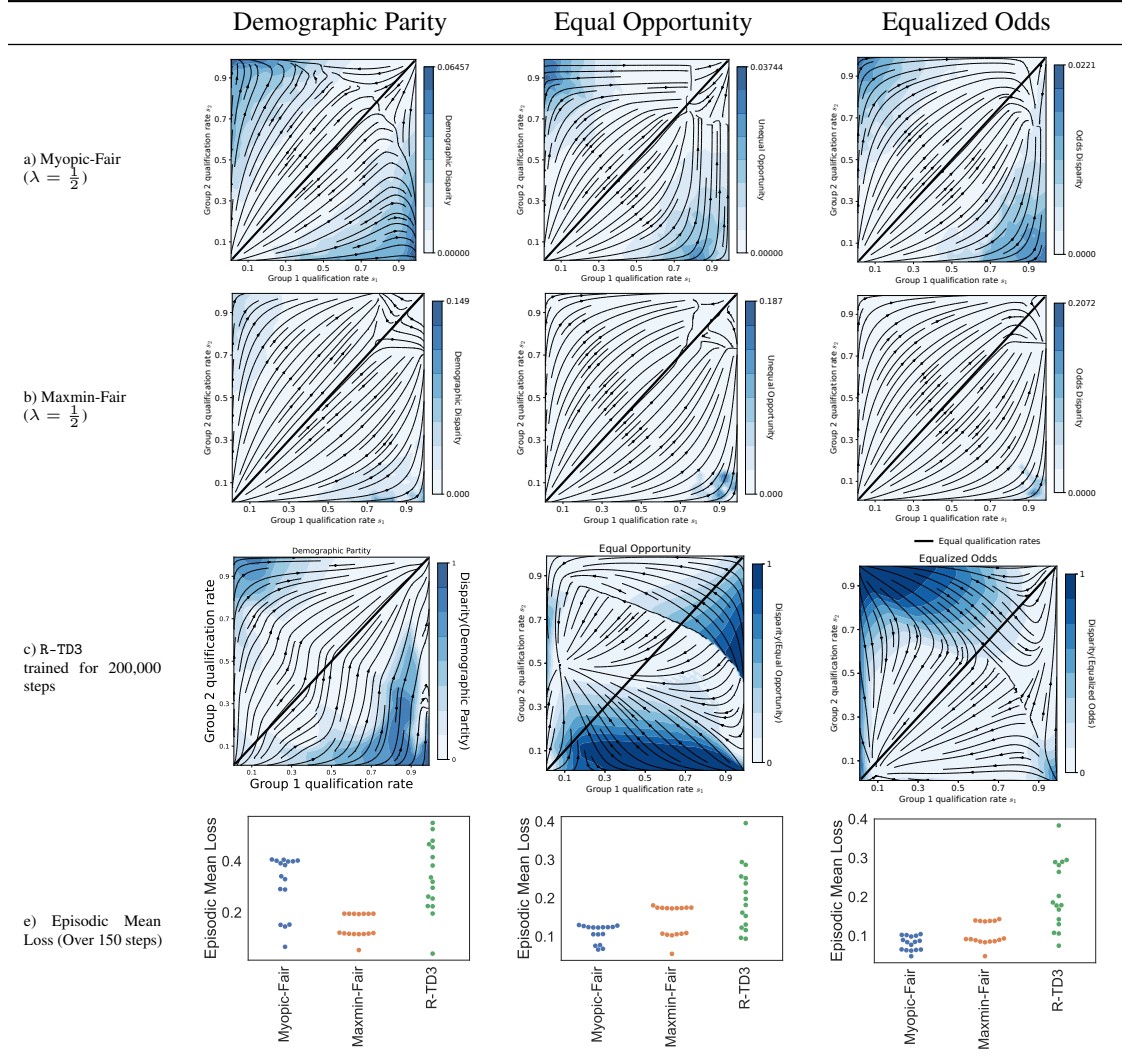

Table 5: A comparison of baseline and R-TD3 policies in a semi-synthetic environment incorporating data synthesized from the UCI "Adult Data Set" (Dua and Graff, 2017), as detailed in Appendix A.2. Policies share the objective of maximizing a weighted combination of the cumulative true positive and true negative fractions of a binary classification task (Section 4), subject to three fairness constraints (Section 5.1) (columns). This figure indicates a negative result, in which the R-TD3 algorithm does not offer clear benefits over the baseline policies. As noted in the main text, there are no theoretical guarantees for the R-TD3 algorithm, and the possibility of such negative results constitute a risk for real-world deployments.

## D.6 Training Curves: `L-UCBFair`

### D.6.1

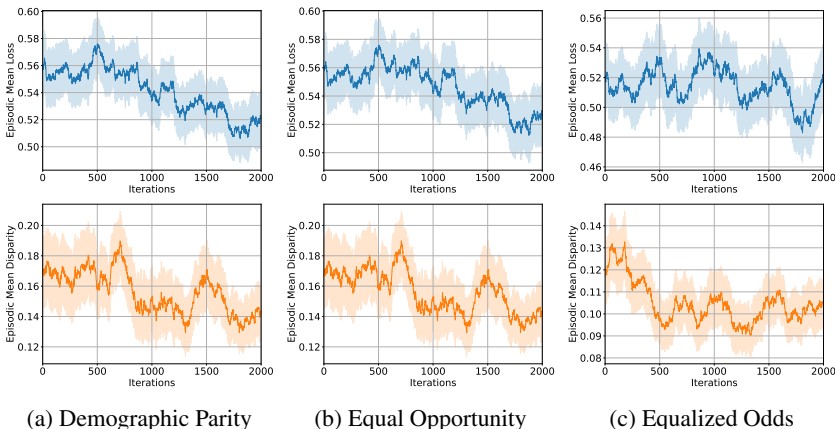

(a) Demographic Parity      (b) Equal Opportunity      (c) Equalized Odds

Figure 7: `L-UCBFair` 20-step sliding mean & std for the setting in Table 2.

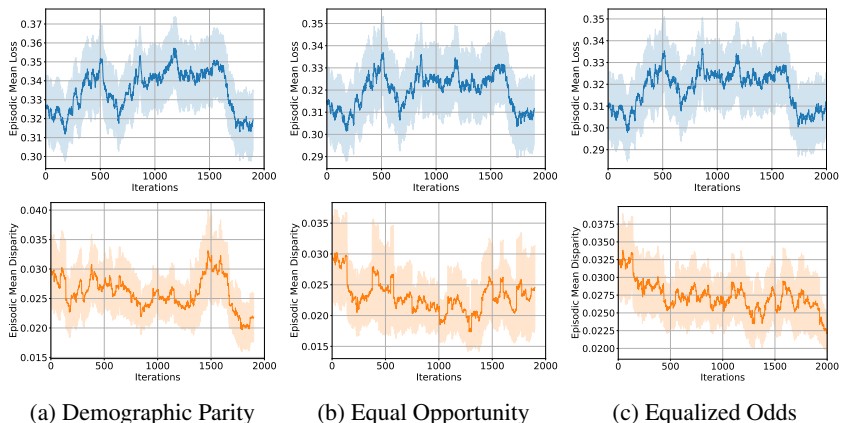

(a) Demographic Parity      (b) Equal Opportunity      (c) Equalized Odds

Figure 8: `L-UCBFair` 20-step sliding mean & std for the setting in Table 3.

## D.7 Training Curves: `R-TD3`

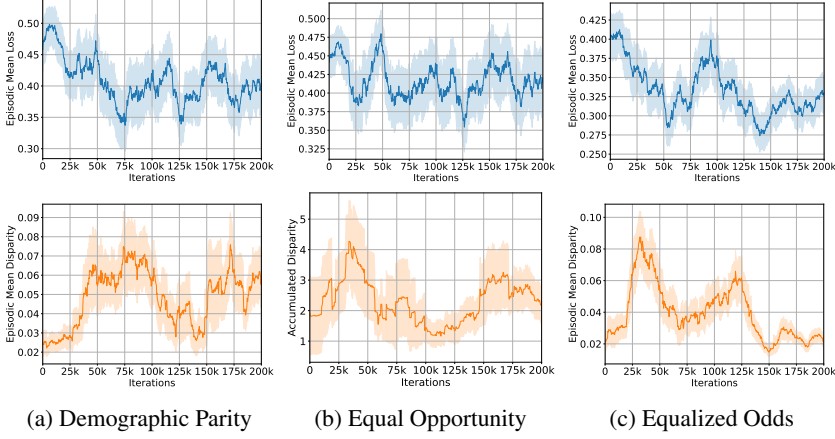

(a) Demographic Parity      (b) Equal Opportunity      (c) Equalized Odds

Figure 9: `R-TD3` 100-step sliding mean & std for the setting in Table 3.

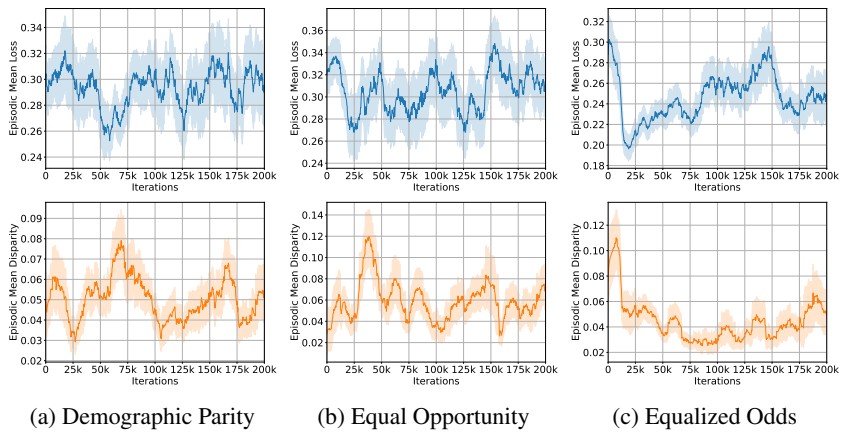

(a) Demographic Parity      (b) Equal Opportunity      (c) Equalized Odds

Figure 10: R-TD3 100-step sliding mean & std for the setting in Table 5.

## D.8 Reduction of Utility

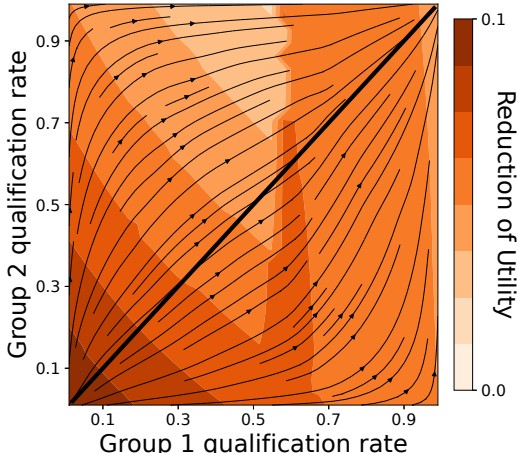

Figure 11: This figure depicts the short-term impact on utility of the `UCBFair` algorithm compared to the greedy, myopic agent that operates without fairness constraints. In this experiment, both algorithms were designed to optimize the fraction of true-positive classifications, but only `UCBFair` was subject to the additional constraint of demographic parity. As the results indicate, the `UCBFair` algorithm experiences a reduction in utility compared to the greedy baseline, but it is able to drive the system towards a state that is preferable in the long term.

