# OpenReview forum: "Long-Term Fairness with Unknown Dynamics"
_NeurIPS.cc/2023/Conference — NeurIPS 2023 poster_

### Official Review · Reviewer_kVGy · 2023-06-11

**Soundness:** 3 good
**Presentation:** 2 fair
**Contribution:** 3 good
**Rating:** 6
**Confidence:** 4

**Summary:**

This paper examines the issue of enduring fairness within a dynamically responsive population, framing it as a reinforcement learning (RL) problem with certain constraints. In this scenario, the state distribution is subject to change dynamically based on the actions of the deployed agent. To address this problem, the authors introduce L-UCBFair and R-TD3. L-UCBFair assumes a simplified linear Markov Decision Process, while R-TD3 incorporates existing deep reinforcement learning methods for a more general setting. According to the numerical experiments conducted, the proposed method consistently surpasses the performance of baseline methods. The RL approach enables an agent to learn how to forego short-term utility to steer the system towards more desirable equilibrium states.

**Strengths:**

**Originality**:
The authors uniquely approach the fairness problem in machine learning within a context where the underlying state distribution dynamically evolves based on the agent's decision. This novel setup extends the static fairness issue found in the existing literature. The dynamic viewpoint introduces the concept of cumulative reward and distortion (cumulative fairness disparity), aptly fitting the method of reinforcement learning. The experiments carried out distinctly highlight the benefits of incorporating a reinforcement learning method, particularly in cases of changing underlying environments.

**Quality and Clarity**:
The motivation behind this work is cogently presented in the introduction, while the related works section explores a wide range of existing literature, discussing both fairness issues in non-stationary settings and safe reinforcement learning. The demonstration of numerical experiments is lucidly laid out.

**Significance**:
The suggested L-UCBFair and R-TD3 methods surpass the performance of baseline methods. The concept of foregoing short-term utility to guide the system towards more optimal equilibrium states is effectively demonstrated. Furthermore, the author presents a thorough theoretical analysis for the proposed L-UCBFair under the linear Markov Decision Process assumption.

**Weaknesses:**

**Weakness 1**:
The depiction of Algorithm 1 could be clearer. There are several terms defined in the paper that aren't elaborated upon, making comprehension more challenging. Enhancing the clarity in Section 3.1.1 could greatly improve overall readability.

**Weakness 2**:
The authors introduce R-TD3 as a broader reinforcement learning method, not reliant on the linear MDP assumption. Could the authors shed more light on the advantages or enhancements that R-TD3 offers in comparison to the foundational method, L-UCBFair?

**Weakness 3**:
In the numerical experiments, the authors have designed synthetic simulation environments based on straightforward datasets. However, in the existing literature, public simulators dedicated to long-term simulation, such as "Fairness is not static: Deeper understanding of long term fairness via simulation studies", are readily available. Given that this has been mentioned in the related works, could the authors assess their proposed L-UCBFair and R-TD3 in these more complex simulation environments? For instance, situations involving lending, attention allocation, or college admissions could provide further insights.

**Weakness 4**:
Reinforcement learning utilizes long-term planning to maximize rewards and diminish distortion, whereas myopic-fair methods – the basis of all baseline methods used in the paper – only address a one-step optimization issue. Given this, the result presented in Figure 1 seems rather self-evident as one would expect a long-term planning-based RL method to surpass more short-sighted methods. Could the authors compare their proposed method with more robust baseline methods that also consider underlying dynamics?

**Questions:**

The issue of long-term fairness can become quite resource-intensive when the agent interacts with the real-world scenarios. Within the existing body of reinforcement learning literature, model-based reinforcement learning methods have been devised to enhance sample efficiency. Could the authors discuss the potential of developing a model-based reinforcement learning approach to tackle the long-term fairness problem addressed in this paper?

Moreover, several queries were raised in the previously mentioned weaknesses section.

**Limitations:**

The authors followed the PaperChecklist by providing a discussion on the current limitations.

---

> ### Author Rebuttal · Authors · 2023-08-10
>
> > Depiction of Algorithm 1 could be clearer. Same for Section 3.1.1.
>
> We express our gratitude for this advice. In the event of acceptance, we intend to enhance the presentation of Algorithm 1 and Section 3.1.1 in the final version. To provide greater clarity, our planned revisions include:
>
> 1. Relocating relatively less crucial parameters, such as $\eta$ and $\chi$, from the algorithm itself. These parameters will be moved to a new section in the Appendix, where we will elaborate on their specifics.
> 2. Segregating the *policy search* details from the primary structure of Algorithm 1. We will introduce a new sub-function dedicated to this aspect. This restructuring aims to maintain the clarity and comprehensibility of the main algorithm. (A potential structure can be provided in .md format.)
> 3. Enhancing the alignment between Section 3.1.1 and Algorithm 1 by incorporating annotations in the algorithm using markers such as #LSVI-UCB, #Adaptive Search Policy, and #Dual Update, similar to the bolded text in Section 3.1.1. Furthermore, when introducing concepts like #LSVI-UCB, #Adaptive Search Policy, and #Dual Update in Section 3.1.1, we will explicitly reference the relevant lines in the algorithm to facilitate readers' better understanding of the content.
>
>
> > The authors introduce R-TD3 as a broader reinforcement learning method, not reliant on the linear MDP assumption. Could the authors shed more light on the advantages or enhancements that R-TD3 offers in comparison to the foundational method, L-UCBFair?
>
> The major advantage of R-TD3 over L-UCBFair is that state-of-the-art deep learning techniques may be used much like R-TD3 (i.e., with a Lagrangian objective subject to scheduled dual variable), potentially providing superior performance, albeit without strong theoretical guarantees. Unlike L-UCBFair, which relies on the linear MDP assumption and thus only trains the weights of the fully connected layer, R-TD3 does not have a requirement for a specific network structure. This makes it suitable for larger and more efficient neural networks, leading to powerful real-world applications involving complicated environments. However, unlike L-UCBFair, R-TD3, as a deep learning framework, does not offer regret guarantees.
>
> > … public simulators dedicated to long-term simulation such as “Fairness is not static: Deeper understanding of long term fairness via simulation studies”, are readily available…, could the authors assess their proposed L-UCBFair and R-TD3 in these more complex simulation environments? For instance, situations involving lending, attention allocation, or college admissions could provide further insights.
>
> While the cited package implements the model proposed by Liu’s “Delayed Impact” paper, it uses discrete action spaces (e.g., only seven possible credit score thresholds), in keeping with the capabilities of more established algorithms. To more appropriately demonstrate our work in addressing continuous action spaces, we required a different experimental environment. We agree that additional experimental evaluation may provide further insights, and hope that future work futhers both algorithmic development and experimental settings for our formulation of long-term fairness with continuous state and action spaces.
>
> > Reinforcement learning utilizes long-term planning to maximize rewards and diminish distortion, whereas myopic-fair methods – the basis of all baseline methods used in the paper – only address a one-step optimization issue. Given this, the result presented in Figure 1 seems rather self-evident as one would expect a long-term planning-based RL method to surpass more short-sighted methods.
>
> We agree that a long-term planning-based RL method should surpass more short-sighted methods in this problem domain (and confirm through experiment that this is so!), and this is why a formulation of long-term fairness, allowing such algorithms to be brought to bear in this domain, is so critical. The state of the art in algorithmic fairness is, sadly, based on myopic classifiers subject to fairness constraints (as listed in Section 1.1). The use of learning techniques to anticipate dynamic population responses to deployed policy and guarantee fairness is nascent.
>
> It may be presumed that the difficulty of treating continuous action spaces, which are common in real-world settings dealing with fairness constraints affecting human populations, has been a primary reason for the delayed treatment of online or RL leaning methods in algorithmic fairness. Our primary contributions are thus an important step for bringing long-term fairness into focus within this research community.
>
> > Could the authors compare their proposed method with more robust baseline methods that also consider underlying dynamics?
>
> We take as a primary assumption that the dynamics with which the population reacts to deployed policy is unknown a priori. When dynamics are known, the field of optimal control may be brought to bear, but different dynamics can require different control policies, and this would have to be analyzed on a case-by-case basis. In general, it is true that knowledge of the dynamics can improve performance in the setting of long-term fairness, but at this early stage, we seek generality.
>
> > Model-based RL can enhance model efficiency. What is potential of model-based methods?
>
> Absolutely, model-based RL can indeed improve performance in specific settings. In particular, when knowledge of the dynamics is known a priori, the class of models can be restricted by this knowledge to improve sample efficiency and performance. Our purpose in this paper has been to provide a general treatment of long-term fairness with minimal assumptions about the specific transition dynamics (i.e., using a linear-MPD assumption), but there is ample room for future work (especially when specialized to specific settings) to adopt model-based RL or approaches deriving from control (e.g., data-driven control).

---

### Official Review · Reviewer_iPFQ · 2023-07-02

**Soundness:** 2 fair
**Presentation:** 3 good
**Contribution:** 2 fair
**Rating:** 5
**Confidence:** 3

**Summary:**

This paper considers a binary classification task with long-term fairness. The authors formulate the problem as a constrained MDP where the classifier is an action, the distribution is the state, the distribution will shift reacting to an action. The problem aims to optimize the expected long-term reward under the constraints for long-term fairness in expectation. They develop L-UCBFair which is a model-free algorithm to guarantee long-term fairness with continuous state and action spaces.  Under the linear MDP assumption, they prove sublunar regret and disparity with high probability.  Finally, experiment results are given to demonstrate the performance of the proposed algorithm.

**Strengths:**

I believe the paper has the following strong points.
+ The paper formulates fairness constrained classification problem as a constrained MDP, which is new to me.
+ The paper applies linear MDP algorithms and TD3 to the considered problem and prove the regret bound with linear MDP assumption.

**Weaknesses:**

The paper can still be improved in the following aspects.
- The paper seems to directly apply LSVI-UCB and TD3 algorithms to solve the considered constrained MDP problem. It would be much better if the authors can discuss how the algorithm design and proof techniques are different from original works in (Jin et al. (2020)) and  Fujimoto et al. (2018).
- The paper formulates the fairness-constrained classification problem as a MDP problem, but does not give enough motivation.
- It usually requires a lot of exploration for the constrained RL to achieve low disparity, so I am afraid that the fairness violation can be very high at early stages.
- The paper is limit in addressing the complexity problem by using constrained RL for this problem.  The action space (classifier) can be very large, which can cause a very large complexity.

**Questions:**

1. The paper considers a dynamic where the new distribution is affected by the previous distribution and classifier. Can the authors give some concrete motivation examples for such distribution shift?
2. Can the authors discuss the challenges of solving the MDP with fairness constraints?  What are the differences comparing with standard constrained MDP problems?
3. How does the complexity of proposed algorithm rely on the space of state and action?

**Limitations:**

The limitations are listed by the authors.

---

> ### Author Rebuttal · Authors · 2023-08-10
>
> > The paper seems to directly apply LSVI-UCB and TD3 algorithms... It would be much better if the authors can discuss how the algorithm design and proof techniques are different from original works in (Jin et al. (2020)) and Fujimoto et al. (2018).
>
> The L-UCBFair differs from LSVI-UCB (Jin et al. 2020) in important respects:
>
> 1. LSVI-UCB constitutes an RL framework devoid of constraints. The objective in Jin et. al.’s paper is to maximize cumulative utility, whereas ours is to optimize utility subject to a fairness constraint. Due to this constraint, we must monitor the value function associated with cumulative fairness violations (i.e., “distortion” in our paper), and derive bounds on  the associated regret. Deviating from the cited work, we employ a primal-dual approach and formulate a policy that takes both the objective and fairness constraint into account (refer to algorithm 1, SM(...)). The resulting proof establishes regret and distortion bounds at O(H^2√d^3K), both with high probability.
> 2. LSVI-UCB exclusively applies to discrete action spaces. Without the ability to utilize continuously varying policies, It may be impossible to satisfy fairness conditions, just as it may be impossible to find the roots of an arbitrary polynomial when restricted to integer-valued inputs. For dealing with fairness, our extension of LSVI-UCB with an adaptive search policy is necessary.
>
> Similarly, R-TD3 differs in an important way from direct application of TD3 (Fujimoto et al. 2018):
> 1. The objective on which we use TD3 is a Lagrangian relaxation of the constrained optimization problem with a schedule for the dual variable, where the schedule introduces an additional hyperparameter.
>
> > The paper formulates the fairness-constrained classification problem as a MDP problem, but does not give enough motivation… The paper considers a dynamic where the new distribution is affected by the previous distribution and classifier. Can the authors give some concrete motivation examples for such a distribution shift?
>
> *Regarding motivation and examples:* Deployment bias is a frequent occurrence in machine learning systems where the model interacts frequently with users and the model's outputs have a high impact on users' well-being (e.g., financial decision-making) and preferences (e.g., recommendation system).
>
> Imagine two distinct user groups in the loan approval setting, namely Group $A$ and Group $B$. Let's assume that the deployed model has a much higher approval rate for the applicants from Group $A$, and that Group $A$ is in a better position to put loans to productive use through investment. Over time, as users from $A$ receive more financial support, they can further improve their financial position (e.g., by receiving education, by starting their own business) and are likely going to become even more qualified for loans in the future. On the other hand, users from Group $B$ lacked the opportunity to grow financially and were trapped in a low socioeconomic status, hurting chances for future loan applications. In this example, the policy (higher approval rates for Group $A$) and the state (Group $A$’s ability to use loans productively) jointly led to a new state where Group $A$ is even more able to use loans productively than Group $B$, which has become trapped in a state with few prospects and little opportunity for advancement.
>
> *Regarding limitations:* The MDP formulation is quite general; the only restriction it imposes in our use case is an assumption of observability (i.e., that no state variables exist other than the distribution itself). While this assumption is not strictly necessary, it does simplify the formulation and eliminates the need for considering additional state information.
>
> > It usually requires a lot of exploration for the constrained RL to achieve low disparity, so I am afraid that the fairness violation can be very high at early stages.
>
> This fear is not unfounded, however the violation of fairness and increased loss at early stages may in fact be necessary when short-term incentives and long-term benefit may be misaligned in practice (such that myopic policies may steer the system toward undesirable equilibria, as in Fig 1 (b)). This aspect of our work is highlighted in the introduction and further demonstrated in experiments: To guide the system towards more desirable equilibria, an RL formulation of long-term fairness is imperative, and an online setting is unavoidable (See response to first weakness pointed out by Reviewer oXaL).
>
> > Can the authors discuss the challenges of solving the MDP with fairness constraints? What are the differences compared with standard constrained MDP problems?
>
> For verisimilitude, the key complications to a standard constrained MPD problem introduced by our consideration of fairness are
> 1. An assumption that the MDP transitions are unknown, necessitating the use of the RL method to derive the policy, given the usual lack of knowledge about real-world dynamics.
> 2. The use of continuous action spaces, which are often necessary to satisfy fairness constraints and are ubiquitous in the real-world (e.g., when actions correspond to policy parameters).
> No prior model-free constrained MDP for continuous action spaces has been theoretically equipped with regret and distortion bounds. Our work breaks new ground by introducing the first algorithm of its kind, providing such guarantees.
>
> > The action space (classifier) can be very large, which can cause large complexity. …. How does the complexity of the proposed algorithm rely on the space of state and action?
>
> State: For state space, our memory requirements are linear in the dimensionality of the learned embedding space ($\phi$ in the paper).
>
> Action: For action space, our memory requirements are linear in the number of action space regions ($M$ in the paper). As a trade-off, $M$ affects $\epsilon_I$ (Theorem B.4), and thus regret bounds for the algorithm (Equation 13 and line 526).

---

> > ### Comment · Reviewer_iPFQ · 2023-08-14
> > **Thank you for rebuttals**
> >
> > I thank the authors for answering my questions. I acknowledge reading the rebuttal and have further questions.
> >
> > **Difference from LSVI_UCB and TD3.**  I am convinced that this work is different from LSVI_UCB and TD3 since they do not consider constraints.
> >
> > **Motivation of RL formulation.** The example of loan approval is a good motivation for this problem, but the limitations about the observability should be discussed in the paper.
> >
> > **Fairness violation.** I understand that the constant violation is not avoidable for the online setting.
> >
> > **Difference from constrained MDP.** I understand that the analysis generalizes the constrained MDP with finite action space (Ghosh et al. (2022)) to continuous action space. The issue introduced by continuous action is solved by discretizing the action space which is shown in the proofs of Lemma B.10, B.11, B.12.  Please response to me if I miss other technical challenges.
> >
> > **Complexity** I understand that the regret bound depends on the dimensionality of the embedding instead of directly on the size of the action-state space. However, dimensionality of the embedding may also implicitly increases with the size of the action-state space. The authors may need to discuss this scalability issue, especially for this setting where the state includes the full distribution information.

---

> > > ### Author Response · Authors · 2023-08-14
> > >
> > > We appreciate the reviewer's engagement with our rebuttal. Below, we provide the answers to the remaining questions.
> > >
> > > > **Motivation of RL formulation.** The example of loan approval is a good motivation for this problem, but the limitations about the observability should be discussed in the paper.
> > >
> > > Thank you for additional feedback. At present, we state on lines 123-125 that “We assume $s_\tau$ is fully observable at time $\tau$. In practice, $s_\tau$ must be approximated from finitely many empirical samples, though this caveat introduces well-understood errors that vanish in the limit of infinitely many samples.”
> > >
> > > We will be sure to mention this limitation again in the “Limitation” paragraph at line 309.
> > >
> > > > **Difference from constrained MDP.** I understand that the analysis generalizes the constrained MDP with finite action space (Ghosh et al. (2022)) to continuous action space. The issue introduced by continuous action is solved by discretizing the action space which is shown in the proofs of Lemma B.10, B.11, B.12. Please respond to me if I miss other technical challenges.
> > >
> > > Indeed, this is our solution. However, besides the proofs of Lemma B.10, B.11, B.12,
> > >
> > > 1. We also analyze the distinction between $\left|Q^k_{j, h}(s, a)-Q^k_{j, h}(s, I(a))\right|$, imposing $\epsilon_I$ in Lemma B.14. With this Lemma we can then derive the first inequality of proof of Lemma B.10.
> > >
> > > 2. Lemma B.3 and Theorem B.4 are also needed for the adaptive searching policy.
> > >
> > > The proofs of Lemma B.5, B.7, and Theorem 3.3, likewise, exhibit technical novelty.
> > >
> > > > **Complexity.** I understand that the regret bound depends on the dimensionality of the embedding instead of directly on the size of the action-state space. However, dimensionality of the embedding may also implicitly increase with the size of the action-state space. The authors may need to discuss this scalability issue, especially for this setting where the state includes the full distribution information.
> > >
> > > In general, it is true that larger joint state-action spaces admit more complexity, but this is not necessarily so: Koopman operator theory (which ensures the existence of a linear operator that describes the evolution of observables in a dynamical system) in general can require infinite dimensions independent of the dimension of the underlying state of a dynamical system. This is not an issue of scalability with regard to dimension, but with respect to complexity. The best way to measure this complexity is by the minimal necessary dimension of $\phi$, which can be independent of the dimensionality of the joint state-action space.

---

> > > > ### Comment · Reviewer_iPFQ · 2023-08-15
> > > > **Thank you for your response!**
> > > >
> > > > Thank you for your response!
> > > >
> > > > The proofs of Lemma B.5, B.7 and Theorem 3.3 extends the counterparts in (Ghosh et al. (2022)) by applying Lemma B.10, 11, 12, so I feel that main technical challenges are in Lemma 10, 11, 12.
> > > >
> > > > It seems that Lemma B.3 and Theorem B.4 are presented without being proved. Are they from other literatures?

---

> > > > > ### Author Response · Authors · 2023-08-16
> > > > > **Thank you for your response**
> > > > >
> > > > > We appreciate your time and attention in continuing the discussion.
> > > > >
> > > > > > It seems that Lemma B.3 and Theorem B.4 are presented without being proved. Are they from other literature?
> > > > >
> > > > > We’ll add the proof of Lemma B.3 and Theorem B.4 to our paper. Here, we present the proofs.
> > > > >
> > > > > **Definition B.2.** Given a set of distinct actions $I=\\{ I_0, \cdots, I_M \\} \subset \mathcal{A}$ , where $\mathcal{A}$ is a closed set in Euclidean space, define $\mathcal{I}_i=\\{a:\left\\|a-I_i\right\\|_2 \leq\left\\|a-I_j\right\\|_2, \forall j \neq i\\}$ as the subset of actions closer to $I_i$ than to $I_j$, i.e., the Voronoi region corresponding to locus $I_i$, with tie-breaking imposed by the order of indices $i$.
> > > > >
> > > > > Also define the locus function $I(a)=\min_i \arg \min_{I_i}\left\\|a-I_i\right\\|_2$.
> > > > >
> > > > > **Lemma B.3.** The Voronoi partitioning described above satisfies
> > > > > 1. $\mathcal{I}_i \cap \mathcal{I}_j = \varnothing, \forall i \neq j$
> > > > > 2. $\cup_{i=1}^M \mathcal{I}_i=$ $\mathcal{A}$.
> > > > >
> > > > > Proof. We will begin by proving $\mathcal{I}_i \cap \mathcal{I}_j=\varnothing$ for all $i \neq j$. To establish this, we will assume the contrary, that there exist indices $i$ and $j$ such that $\mathcal{I}_i \cap \mathcal{I}_j \neq \varnothing$. Without loss of generality, assume $i>j$. We will denote an arbitrary action within the interaction of $\mathcal{I}_i$ and $\mathcal{I}_j$ as $a^{\prime} \in \mathcal{A}$.
> > > > >
> > > > > Since $a^{\prime} \in \mathcal{I}_i$, according to the given Definition B.2, we can infer that $\left\\|a^{\prime}-I_i\right\\|_2<$ $\left\\|a^{\prime}-I_j\right\\|_2$ (since $i>j$ ). However, this assertion contradicts the fact that $a^{\prime} \in \mathcal{I}_j$, which implies $\left\\|a^{\prime}-I_j\right\\|_2 \leq\left\\|a^{\prime}-I_i\right\\|_2$. Therefore, $\mathcal{I}_i \cap \mathcal{I}_j=\varnothing$ for all $i \neq j$.
> > > > >
> > > > > We then proof $\cup_{i=1}^M \mathcal{I}_i=\mathcal{A}$.
> > > > >
> > > > > Since $\mathcal{A}$ is a closed set, for any $a \in \mathcal{A}$, there must be a $i \\in\\{1,2, \cdots, M\\}$, such that $d_{a, i} = \\|a-I_i\\|_2 \\leq \\|a-I_j \\|_2=d_{a, j}, \\forall j$.
> > > > >
> > > > > If $d_{a, i}<d_{a, j}$ strictly holds for all $j$, then $a \in \mathcal{I}_i$.
> > > > >
> > > > > Otherwise define a set $\\mathcal{J}=\\left\\{j \\mid d_{a, j}=d_{a, i}\\right\\}$, then $a \\in \\mathcal{I}_{j'}$,
> > > > >
> > > > > where $ j'=\\arg\\min_{j \\in \\mathcal{J} } j$.
> > > > >
> > > > > **Theorem B.4.** If the number $M$ of distinct loci or regions partitioning $\mathcal{A}$ is sufficiently large, there exists a set of loci $I$ such that $\\forall a \\in \\mathcal{I}_i, i \\in M,\\left\\|a-I_i\\right\\|_2 \\leq \\epsilon_I$.
> > > > >
> > > > > Proof. Since $\mathcal{A}$ is closed in Euclidean space, denote $N$ the dimension, $d = \sup_{a, a’ \in \mathcal{A}} \\| a - a’ \\|_2$.
> > > > >
> > > > > Define an Orthonormal basis $\\{e_1, e_2, \cdots, e_N\\}$. Randomly choose a point $a \in \mathcal{A}$, Set it as $I_0$, then we form a set of loci $I = \\{I_0 + \sum_{i=1}^{N} \epsilon k_i e_i | I_0 + \sum_{i=1}^{N} \epsilon k_i e_i \in \mathcal{A}, k_i \in \mathcal{Z}, - \lceil \frac{d}{\epsilon} \rceil \leq k_i \leq \lceil \frac{d}{\epsilon} \rceil \\}$. We know that $|I| \leq \left(2 \lceil \frac{d}{\epsilon} \rceil\right)^{N} $. It's not hard to verify that $\|a - I_i \|_2 \leq \frac{\epsilon}{2}  \sqrt{2}^{N-1}, \forall a\in\mathcal{I}_i$. Taken $\epsilon = \frac{2 \epsilon_I }{\sqrt{2}^{N-1}}  $ yields the statement.

---

> > > > > > ### Comment · Reviewer_iPFQ · 2023-08-19
> > > > > >
> > > > > > I thank the authors for the proofs of the two lemmas and the plan to add them in the paper.
> > > > > >
> > > > > > I would also recommend the authors to explain the intuition of the regret bound in Theorem 3.3 and how the bound is different from the previous one in the future versions.
> > > > > >
> > > > > > I believe this paper is technically solid, and I increase the score from 4 to 5.

---

> > > > > > > ### Author Response · Authors · 2023-08-21
> > > > > > >
> > > > > > > We would like to express our gratitude to the reviewer for acknowledging our efforts and for adjusting their score. In response to the concerns raised, we have introduced and compared the regret and distortion bounds to existing literature in lines 220 to 222. We will provide a more comprehensive explanation of these bounds in the appendix of our paper.

---

### Official Review · Reviewer_oXaL · 2023-07-07

**Soundness:** 2 fair
**Presentation:** 3 good
**Contribution:** 2 fair
**Rating:** 4
**Confidence:** 3

**Summary:**

This paper introduces a new method to ensure long-term fairness in reinforcement learning. The method is compatible with different utility measures (e.g., the accuracy of the classifier when the goal is to predict a label) and different fairness constraints (e.g., demographic parity or equal opportunity). The method is an online reinforcement learning method, so the agent has to interact with its environment in order to improve the quality of the policy, either in the real world or in simulation. The authors introduce two measures of regret, one related to the primary utility goal and one related to the fairness measure, and present an upper bound for both regrets achieved by the proposed algorithm. The paper presents empirical results using a mixture of synthetic and real data.

**Strengths:**

- This paper tackles the very important problem of ensuring fairness in a sequential decision-making setting. It uses insights from the - reinforcement learning literature to solve a supervised learning fairness problem.
- The proposed method extends existing ones to the continuous setting (i.e., continuous states—e.g., attributes—and continuous actions).
- The paper is very thorough with the mathematical formulation, assumptions, and definitions necessary to fully describe the method.

**Weaknesses:**

- The proposed method is an online RL method, where the learning process occurs while the policy is applied. This is often not realistic in scenarios where we would like to ensure fairness, as intermediate policies could lead to unfair behavior.
- My understanding is that the proposed method is an extension of the method presented by Ghosh et al. (2022) for the continuous setting, where the method itself is fairly similar, and the main new contribution is showing that it is possible to achieve the same bound in this new setting. I believe that this could be better clarified in the paper to highlight the novelty of this work more clearly.
- In the introduction, the authors state that the primary contribution of this work is to consider the problem of long-term fairness as a reinforcement learning problem subject to a constraint. However, previous works have already considered this setting (e.g., the method on which the present work is based).
- I believe the experiments with R-TD3 are not strongly supporting the argument that an algorithm without theoretical guarantees can still be used for long-term fairness. Applying the method in just two scenarios does not present enough empirical evidence to support the claim that the final policy is still fair, and without any theoretical guarantees, this is essential.

**Questions:**

- Could the authors please provide some motivating examples for when an online RL method would be useful/applicable, considering the fact that the policy is updated while being deployed and the regret bounds are only for the final distortion and not the disparity at each timestep (possibly causing intermediate policies to be unfair)?
- The color scheme in Figure 1 is slightly unclear. The range for the demographic disparity in the first two plots is between zero and 0.005 or zero and 0.07 (a very small interval), but for the proposed method results it is between zero and 1. This makes it harder to compare the fairness measures. Could the authors please clarify this?
- Why did the authors select only the Myopic-fair baseline for the second experiment? From the first experiment, it seemed like this wasn’t one of the most competitive baselines, so it would be interesting to see how the other baselines perform in this second scenario as well.

**Limitations:**

The authors discuss the limitations of the work. No major concerns.

---

> ### Author Rebuttal · Authors · 2023-08-10
>
> Thank you for your constructive review. We hope that our rebuttal satisfactorily addresses the perceived weaknesses of the submission and questions you have.
>
> ### Weaknesses
>
> > The proposed method is an online RL method… intermediate policies could lead to unfair behavior.
>
> Unfortunately, there is likely a greater cost to attempting to learn to model the behavior of human populations in response to currently deployed (and potentially naive) algorithmic policies offline (and off-policy), while currently deployed algorithmic policies do not adaptively attempt to control for fairness. In addition, such a model might differ when we change the hypothesis space of the models, or the application contexts, and probably will change when the external environment for how human users interact with the models changes.
>
> Ultimately, if we simultaneously wish to
> * learn to model the how human populations respond to deployed policy,
> * respect fairness,
> and do so in a context-free and data-driven way. Then we fundamentally have an online reinforcement learning problem.
>
> The real-world use of ML is already online: live experiments (intentional or not) are being run on human populations using current ML policies. It is difficult to maintain the position that we should not be seeking to learn from these live experiments and systematically adapting our algorithms.
>
> In addition, we point out that short-term violations of “fairness” may be necessary, in certain settings, to drive the system towards desirable equilibria. Our formulation of long-term fairness takes the approach of constraining cumulative violations of fairness.
>
> > My understanding is that the proposed method is an extension of the method presented by Ghosh et al. (2022) for the continuous setting… the main new contribution is showing that it is possible to achieve the same bound in this new setting. I believe that this could be better clarified…
>
> This understanding is correct; we will adopt this recommendation. While the proof sketch inherits from prior work, the details are novel.  Specifically, as Ghosh et. al. treat a discrete action space, their proof techniques (such as their value function bound, action-value function bound, softmax policy bound, etc.) are not applicable in a continuous setting, and neither is their regret bound (Lemmas 3, 8, 9, 11, 13, 15, 16 of Ghosh et. al.). We solve these issues with the following results:
>
> 1. In Lemma B.10, we derive a distinct bound for the difference between $\bar{V}{h}^{k}(s)$ and $V{h}^{k}(s)$.
> 2. In Lemma B.11, we present novel bounds for softmax policies based on different action-value functions and dual variables.
> 3. Lemma B.12 provides insights into the disparity of $\left|V_{j}^{k} - \widetilde{V}_{j}\right|$.
> 4. We analyze the distinction between $|Q^{k}{j,h}(s, a) - Q^{k}{j,h}(s, I(a))|$ in Lemma B.16.
>
> The proofs of Lemma B.5, B.7, and Theorem 3.3, likewise, exhibit technical novelty.
>
> > …the authors state that the primary contribution of this work is to consider the problem of long-term fairness as a reinforcement learning problem subject to a constraint. However, previous works have already considered this setting (e.g., the method on which the present work is based).
>
> 1. No prior work, to our knowledge, has applied reinforcement learning subject to constraint to the domain of algorithmic fairness, as discussed in Section 1.1 (Dynamics of Fairness in Machine Learning). One of our primary contributions is this formulation and framing of algorithmic fairness.
> 2. Formulating realistic settings for algorithmic fairness presents unique challenges (eg. continuous action spaces) which we address for the first time. No prior model-free constrained MDP for continuous action spaces has been theoretically equipped with regret and distortion bounds.
>
> > I believe the experiments with R-TD3 are not strongly supporting the argument that an algorithm without theoretical guarantees can still be used for long-term fairness. … just two scenarios does not present enough empirical evidence to support the claim that the final policy is still fair, and without any theoretical guarantees, this is essential.
>
> This belief is welcome, and we agree. We are not making the argument that this *specific* algorithm should be used for long-term fairness. Rather, we are claiming that the possibility still exists for a tradeoff between tractability and theoretical guarantees to be exploited, and anticipate additional future work in this direction. In particular, as pointed out in response to Reviewer kVGy, there is ample opportunity for future work to use model-based reinforcement learning or methods from data-driven control when knowledge of transition dynamics is given.
>
> ### Questions
>
> > Could the authors please provide some motivating examples for when an online RL method would be useful/applicable, considering … [the approach may cause] intermediate policies to be unfair?
>
> Our regret bounds apply to cumulative suboptimality. As such, the violation of fairness and increased loss at early stages may in fact be necessary, as short-term incentives and long-term benefit may be misaligned in practice (such that myopic policies may steer the system toward undesirable equilibria, as in Fig 1 (b)). This aspect of our work is highlighted in the introduction and further demonstrated in experiments: To guide the system towards more desirable equilibria, an RL formulation of long-term fairness is imperative.
>
> > The color scheme in Figure 1 is slightly unclear…. Could the authors please clarify this?
>
> We agree that normalizing each figure’s color scale for maximum dynamic range would lead to greater clarity. We have regenerated these figures, providing an example in the rebuttal pdf.
>
> > … it would be interesting to see how the other baselines perform in this second scenario as well.
>
> We give full comparisons to all baselines in the supplementary material. For this setting, see Column 1 of Table 2.

---

### Official Review · Reviewer_nDcm · 2023-07-25

**Soundness:** 3 good
**Presentation:** 3 good
**Contribution:** 2 fair
**Rating:** 6
**Confidence:** 3

**Summary:**

The paper demonstrates that reinforcement learning algorithms can be used to satisfy long term fairness constraints in dynamic environments in which a classifier and population interact while finding high-value equilibria.

The main approach is to treat the problem as a constrained optimization and optimize the Lagrangian function. The authors use a linear UCB-style algorithm to obtain provable regret bounds on both the cumulative value and constraint violations and demonstrate effectiveness of that algorithm and a less-principled-but-still-practical RL alternative using small simulation studies.

Adapting previous work to the interacting classifier population setting requires handling a continuous action space (setting per-group classifier thresholds) - which the authors handle by showing the continuous action space in this setting can be searched in a discretized manner without losing guarantees.

The simulation experiments demonstrate that the RL approaches can find good operating points where more myopic policies can end up exacerbating qualification rate disparities.


**Strengths:**

Significance & Originality: The authors address an important question: How to design decision-making algorithms for dynamic settings where the underlying dynamics are not well known and the designers want to be both fair and encourage positive societal outcomes overall.

Adding fairness considerations to an RL algorithm with a suitably specified reward function is not a groundbreaking idea, but this work offers a concrete algorithm and practical model-free approach for this question that (as far as I can tell) thus far have not been proposed.

Clarity: Paper is very clear and easy to read up to the experiments, setting and notation are clear, as well as the paper’s goals and contributions.


**Weaknesses:**

I found the experiments to be a little bit unclear  in terms of what was being demonstrated with each figure / experiment. What is the role of the fully synthetic vs semi-synthetic experiments? I think the paper could be improved with a clearer exposition of the takeaways from each experiment.

The authors note that real-world experiments would be necessary to really assess impact of this algorithm. I strongly agree with this sentiment and while I understand that truly real-world experiments can be hard to find, the authors could potentially test across a range of dynamic synthetic environments to give a better sense of the generalizability of the proposed approach.


**Questions:**

1. The shading in figures 1a-d suggests that the demographic parity violation can be mapped to the same state space as qualification parity? Is that really true? For any pair x,y on those plots is there a unique value for the demographic parity violation?

2. How would a purely RL-based long-term reward maximizing agent perform as a baseline (with no fairness constraints)? I would think it might also drive the qualification rates up to the top-right of figures 1a-d, in which case can we be sure it’s the fairness constraints that are actually doing the work here and not just the long-term reward optimization in general?


**Limitations:**

Authors addressed the limitations of their experimental setup with synthetic environments. They might also consider discussing what types of societal inequalities can and cannot be addressed by fairness-in-decision-making style constraints.

---

> ### Author Rebuttal · Authors · 2023-08-10
>
> Thank you for your review, your constructive feedback, and your comments.
>
> ### Weaknesses
>
> > I found the experiments to be a little bit unclear…
>
> We will incorporate more succinct takeaways from the experiments in revision.
>
> *What was demonstrated:* Our experiments seek to confirm our central hypothesis: algorithms for long-term fairness may sacrifice short-term incentives for long-term benefits by driving the system to desirable equilibria. More specifically, *in contrast to myopic algorithms, our algorithms are more consistent in achieving desirable states, less fairness violations in equilibrium, and lower mean loss.* We show this with three types of graphical results:
>
> 1. The first, exemplified by Fig 1 (a-d) show the state transitions (i.e., between group qualification rates, which are the x and y axes) driven by different algorithms. Streamlines indicate the next state the corresponding algorithm drives the qualification rates of the two groups to. In these figures, the upper-right corner of the depicted state space, corresponding to higher qualification rates for both groups, is more desirable. Color indicates the degree to which the policy violates fairness in each state, such that lighter color is more desirable (lower fairness violations).
> 2. The second type of graph, exemplified by Fig 1 (e), compares the mean loss sustained by different algorithms during deployment. In general, lower mean loss is possible only when the algorithm succeeds in first achieving a more desirable state.
> 3. The third, exemplified by Fig 3, shows how mean episodic loss and disparity evolve in time; That these quantities decrease in time validates our claim that L-UCB Fair achieves sublinear regrets.
>
> We provide many additional experiments in the supplementary material.
>
> *Synthetic vs semi-synthetic settings:* In general, it is difficult to model population response and distribution shift, and models must strive for verisimilitude while remaining interpretable. The synthetic setting provides an idealized setting in which to evaluate our algorithms, while the semi-synthetic setting adds additional real-world complications and serves as a more robust test of our claims, given that we are unable to ethically or realistically deploy our algorithms on real-world populations.
>
> > …the authors could potentially test across a range of dynamic synthetic environments to give a better sense of the generalizability of the proposed approach.
>
> This is  non-trivial and an active area of ongoing research: We are unaware of existing implementations of synthetic environments that model policy-induced distribution shifts and combine continuous state and action spaces. While off-the-shelf synthetic environments (e.g., for continuous control) exist for reinforcement learning in general, for our setting, which combines fairness and distribution shift, it remains an open research question how to appropriately model how human populations respond to algorithmic policies.
>
> ### Questions
>
> > …the demographic parity violation can be mapped to the same state space as qualification parity? ...For any pair x,y on those plots is there a unique value for the demographic parity violation?
>
> This mapping from state to disparity is induced by the policy: Because the policy maps state to action, and disparity is a function of state and action jointly, each policy induces a mapping from state to disparity. The mapping is one-way however: it is not true in general that each state will result in a unique value of disparity.
>
> > How would a purely RL-based long-term reward maximizing agent perform as a baseline (with no fairness constraints)? … can we be sure it’s the fairness constraints that are actually doing the work here and not just the long-term reward optimization in general?
>
> Please find the results of running TD3 on (i.e., R-TD3 with lambda fixed to 0) in the rebuttal PDF. In general, it is true that long-term utility may be aligned with fairness, but it is possible that long-term utility alone may work against long-term fairness, in which case the schedule of ever increasing $\lambda$ is important. For a high-level example, consider college admissions, for which only a finite number of applicants can be admitted. When one “elite” group is privileged to be objectively better prepared for a college’s target curriculum, it is easy to imagine how long-term utility measured only by post-admission success rates would exclude disadvantaged groups, absent fairness interventions.
>
> In either case, the point remains that myopically “fair” baselines can do active harm, while our formulation of long-term fairness addresses both cumulative utility *and* cumulative fairness violations. Moreover, ours is the first work we are aware of that addresses long-term utility subject to policy-driven distribution shift in human populations that we are aware of.
>
> ### Limitations
>
> > Authors … might also consider discussing what types of societal inequalities can and cannot be addressed by fairness-in-decision-making style constraints.
>
> An inherent strength of our formulation lies in its broad generality: Any real-valued function of state (i.e., distribution) and action (i.e., policy), jointly, is a valid measure of fairness in our work, though L-UCBFair assumes that there exists some (possibly high, but finite dimensional) space for which the measure can be represented as a linear map (Assumption 3.1), and R-TD3 similarly requires that the map is (implicitly) “learnable” by the chosen architecture.
>
> Nonetheless, we share the concern that some socially consequential settings are too sensitive to risk to online learning algorithms: Stochastic sampling and exploration may create irreparable or intolerable harm. This is what we mean when we say in the paragraph “Limitations” (line 309) that “violations of fairness or decreased utility may be difficult to justify to affected populations and stakeholders”.

---

> > ### Comment · Reviewer_nDcm · 2023-08-15
> >
> > I acknowledge reading the rebuttal. Thank you for addressing my comments.
> >
> > I appreciate the additional result with TD3 (lambda=0). It seems from that new result that in the simulated settings used for the experiments here, simply optimizing for long-term reward is sufficient to move towards high-fairness results, but in practice the unconstrained approach has a higher cumulative disparity than the constrained setting.
> > I think the synthetic demonstrations would be a little bit stronger if they included a setting where fairness and long-term reward were mis-aligned (as described by the authors in the rebuttal, but not used in the experimental demonstrations), but I don't think this is critical.

---

> > > ### Author Response · Authors · 2023-08-16
> > >
> > > We appreciate your time and attention in reviewing and addressing our rebuttal.
> > >
> > > >  I think the synthetic demonstrations would be a little bit stronger if they included a setting where fairness and long-term reward were mis-aligned (as described by the authors in the rebuttal, but not used in the experimental demonstrations), but I don't think this is critical.
> > >
> > > We agree that additional synthetic environments with natural misalignment between long term fairness and utility would allow further useful characterization of the potential consequences of the algorithms we explore. We hope to target such environments in future work.
> > >
> > > We would greatly value your reconsideration of the rating, if critical concerns have been alleviated.

---

### Author Rebuttal · Authors · 2023-08-10

We thank our reviewers for their constructive feedback and questions, as well as pointing out ways in which we could improve our submission (e.g., by highlighting the novelty of our proofs and improving our figures in subtle ways, which we have done).

With the attached PDF, we have regenerated Figure 1 c) as well as Figure 2 a1) and a2) to highlight that the instantaneous disparity achieved by L-UCBFair in these experiments was bounded below 0.32 in violation of Demographic Parity (Table 1).

To address common concerns among our reviewers, we would like to point out that our experimental setup was necessarily non-trivial, as we sought to model continuous state and action spaces. To our knowledge, no existing experimental settings implement continuous state and action spaces specific to the domain of algorithmic fairness, and our evaluation of our methods was not easily extended to other settings without significant effort. We have no doubt that additional synthetic environments will be proposed and implemented in the course of ongoing investigations into the line of inquiry we open with our formulation of long-term fairness.

Additionally, we would like to reiterate that short-term violations of utility, meaning both decreasing loss and increasing fairness, are expected and often necessary in the settings we explore. As stated in the paper, a central motivation of our work is the necessity of sacrificing short-term incentives in order to drive the classifier-population system to more desirable outcomes. We show in our experiments (e.g., Figure 1 a), b), and Figure 2 a)) that myopic fairness constraints, which prioritize short-term fairness, can lead to worse outcomes in the long-run.

Finally, we emphasize that our formulation of long-term fairness as an online reinforcement learning problem is in many ways unavoidable. In particular, while there are risks associated with online learning, the alternative in reality, where interacting with human populations is a slow and costly proposition, is not “offline” learning, but online non-learning, typified by myopic policies.

We appreciate your attention to our submission and the chance to respond to each of our reviewers individually.

---

### Decision · Program_Chairs · 2023-09-21

**Decision:**

Accept (poster)

**Comment:**

My recommendation is to accept the paper.

The authors clearly show that driving dynamical systems toward fair equilibria is an important problem that is not addressed by static or myopic fairness criteria. There is innovation in both problem framing and technical approach (especially finding an effective way to discretize the continuous action space). Reviewer concerns were well-addressed in the rebuttal period. The unconstrained TD3 results should be added to the camera ready to show distinctness between constrained and unconstrained algorithms in real scenarios.